# FREED++: Improving RL Agents for Fragment-Based Molecule Generation by Thorough Reproduction

**Alexander Telepov**[*a], **Artem Tsypin**[a], **Kuzma Khrabrov**[a], **Sergey Yakukhnov**[c],
**Pavel Strashnov**[a], **Petr Zhilyaev**[d], **Egor Rumiantsev**[a], **Daniel Ezhov**[c], **Manvel Avetisian**[a],
**Olga Popova**[a], **Artur Kadurin**[*a,b]

[*] *Corresponding authors. Contacts: Telepov@airi.net, Kadurin@airi.net*

[a] *AIRI, Kutuzovskiy prospect house 32 building K.1, Moscow, 121170, Russia.*

[b] *Kuban State University, Stavropolskaya Street, 149, Krasnodar 350040, Russia.*

[c] *Sirius University of Science and Technology, Olimpiyskiy ave. b.1, Sirius, Krasnodar 354340, Russia.*

[d] *Independent researcher.*

**Reviewed on OpenReview:** *https://openreview.net/forum?id=YVPb6tyRJu*

## Abstract

A rational design of new therapeutic drugs aims to find a molecular structure with desired biological functionality, e.g., an ability to activate or suppress a specific protein via binding to it. Molecular docking is a common technique for evaluating protein-molecule interactions. Recently, Reinforcement Learning (RL) has emerged as a promising approach to generating molecules with the docking score (DS) as a reward. In this work, we reproduce, scrutinize and improve the recent RL model for molecule generation called FREED (Yang et al., 2021). Extensive evaluation of the proposed method reveals several limitations and challenges despite the outstanding results reported for three target proteins. Our contributions include fixing numerous implementation bugs and simplifying the model while increasing its quality, significantly extending experiments, and conducting an accurate comparison with current state-of-the-art methods for protein-conditioned molecule generation. We show that the resulting fixed model is capable of producing molecules with superior docking scores compared to alternative approaches.

## 1 Introduction

The traditional drug discovery process is notoriously long and expensive. Modern drug discovery pipelines utilize computational methods to decrease the amount of *in vitro* experiments reducing the time and cost of the whole process. One of the crucial early steps of computer-aided drug discovery (CADD) is the virtual screening(VS) (Lyne, 2002) of vast databases of chemical compounds. VS identifies potential drug candidates possessing desired chemical properties (e.g., stability, low toxicity, synthetic accessibility, etc.). Despite being cheaper and faster than the traditional wet-lab approach, VS has another major drawback: it is limited to the databases of already known drug candidates and is not suitable for *de novo* drug design. Moreover, it is estimated that there are more than $10^{60}$ (Reymond & Awale, 2012) potential drug-like molecules, making it impossible to screen all candidates even if such a database was available.

Recently, deep learning (DL) has been used to overcome the limitations of traditional approaches. Early works focus on speeding up the virtual screening process by using neural networks (Shoichet, 2004; Dahl et al., 2014; Ma et al., 2015; Wallach et al., 2015; Ramsundar et al., 2015; Unterthiner et al., 2014; Gawehn et al., 2015; Mayr et al., 2016; Baskin et al., 2016; Koutsoukas et al., 2017). However, despite the existence of a large number of chemical databases, including drug-like molecules (Gaulton et al., 2012; Irwin et al., 2012; Polykovskiy et al., 2020), pre-computed chemical properties (Chmiela et al., 2017; Isert et al., 2022; Khrabrov et al., 2022; Eastman et al., 2023), and protein-ligand pairs (Francoeur et al., 2020; Hu et al.,

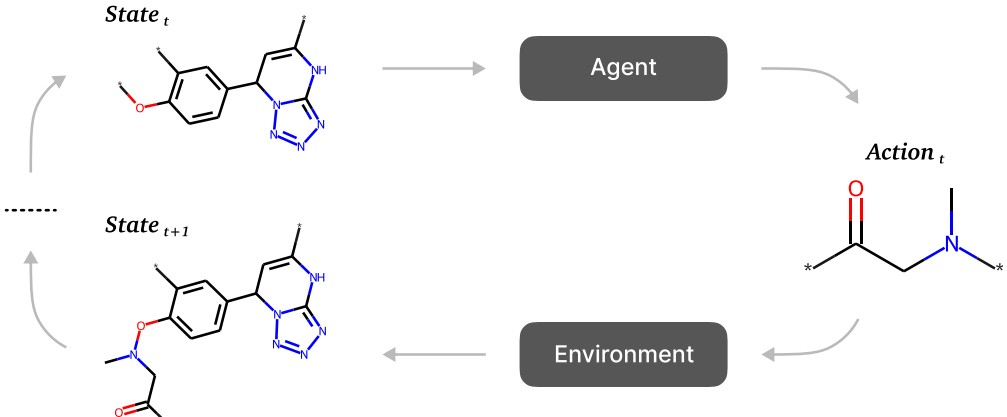

Figure 1: An overview of RL-based sequential generation methods. The agent takes the current state (section 3.2) and selects an action (section 3.3). The action is usually a molecular fragment appended to the current state. Individual atoms or atom bonds are considered trivial cases of fragments. The transition dynamic is straightforward: the new state is assembled from the previous state by attaching a new fragment to it. Note that in some frameworks (Zhou et al., 2019; Jeon & Kim, 2020) a removal of the fragment is also considered an action. The specific task defines the reward. Some examples include cLogP, QED, and various binding affinity proxies. If $J$ is the optimization objective, the reward can be chosen as $r_{t+1} = J(s_{t+1}) - J(s_t)$ or $r_{t+1} = \begin{cases} 0 & \text{if } s_{t+1} \text{ is non-terminal;} \\ J(s_{t+1}) & \text{if } s_{t+1} \text{ is terminal.} \end{cases}$

2005), the size of the drug-like molecule space suggests that generative and other semi-supervised methods may be more promising.

Generative models are trained to map drug-like molecules into a low-dimensional hidden space and restore the original molecules from this low-dimensional representation. This approach allowed the sampling of new drug candidates by the perturbation of low-dimensional representations of existing molecules. Generative models can operate on several different molecule representations, such as SMILES (Gómez-Bombarelli et al., 2018), 2D graphs (Jin et al., 2018; De Cao & Kipf, 2018), hyper-graphs (Kajino, 2019), and 3D structures (Gebauer et al., 2019; Simm et al., 2021). Guacamol (Brown et al., 2019) provides a benchmark for such models.

Despite several notable achievements (Zhavoronkov et al., 2019), such models do not consider the target protein the molecule binds to. This is suboptimal, as the therapeutic effect of the molecule is largely determined by its binding affinity to the target protein. Various groups have proposed an approach to the protein-conditioned generation of molecules based on an autoregressive generative model that sequentially assembles molecules by adding atoms and atom bonds Li et al. (2021); Drotár et al. (2021); Peng et al. (2022); Liu et al. (2022). Diffusion models (Hoogeboom et al., 2022; Igashov et al., 2022; Schneuing et al., 2023; Guan et al., 2023) have been proposed as an alternative to autoregressive sequential generation to speed up the generation process significantly. Both autoregressive and diffusion generation require a dataset of protein-ligand pairs, and the amount of training data limits their quality.

An alternative approach to protein-conditioned molecule generation is to use Reinforcement Learning (RL). RL-based approaches can be roughly divided into two categories. The first category, briefly described in Fig. 1, is the sequential generation (You et al., 2018; Zhou et al., 2019; Jeon & Kim, 2020; Yang et al., 2021). Such models do not require training datasets and can utilize desired molecular properties such as docking score, drug-likeness (Lipinski et al., 1997), epitope score (Shashkova et al., 2022), or chemical constraints into the training objective. In contrast, most of graph generative models such as VAEs, Diffusion models or GANs cannot directly incorporate such properties into their training objective. However, different strategies, such as active learning or evolutionary algorithms, can be used to optimize desired properties (Segler et al., 2018; Schneuing et al., 2023). The second category (Popova et al., 2018; Blaschke et al., 2020; Thomas et al., 2021; Ghugare et al., 2023; Mazuz et al., 2023) uses RL to fine-tune pre-trained generative models on large

datasets to produce molecules with desired properties, such as cLogP and QED. Unlike the first category, these models require two-step training and a dataset to pre-train the generative model on.

The binding affinity of a molecule to the target protein is a perfect reward for protein-conditioned molecule generation. However, since the real binding affinity is intractable, recent research has utilized its proxy, Docking Score (DS), as a reward in RL setting (Jeon & Kim, 2020; Cieplinski et al., 2020; Thomas et al., 2021; Yang et al., 2021). In this work, we focus on FREED pipeline (Yang et al., 2021), which sequentially generates molecules using fragments as building blocks. This allows to ensure the validity of generated molecules that have high DS for various target proteins.

We strongly believe in the approach used in FREED, so in this work, we conducted a thorough analysis of its implementation and experimental setup. Our findings are as follows: i) the implementation contains multiple bugs, ii) the evaluation setup is inconsistent between the proposed model and baselines, iii) the amount of target proteins selected for model evaluation is insufficient, and iv) the model is overcomplicated and lacks ablations.

In order to address these limitations, we meticulously inspected the code, fixed the bugs, conducted ablation studies, and simplified the proposed model. We have named the resulting fixed and simplified model FREED++. Moreover, we have substantially broadened the scope of our experiments. To avoid cherry-picking of target proteins, we have tested FREED++ on a large number of proteins from the DUD-E database (Mysinger et al., 2012), and additionally, on the ubiquitin-specific protease 7 (USP7) protein (Leger et al., 2020a). We have experimented with fragment libraries and additional structural constraints to provide further insight into the model. Finally, we have fixed the evaluation protocol and compared FREED++ with alternative approaches to accurately compare it with current SOTA protein-conditioned molecule generation methods. We find that FREED++ exhibits great generalization ability while producing valid molecules with docking scores superior to those of alternative approaches. The code is accessible by link [1].

## 2 Related work

**Current state of protein-conditioned molecule generation** Our work is concerned with protein-conditioned molecule generation. This area of research has received increased attention lately. REIN-VENT (Olivecrona et al., 2017; Thomas et al., 2021), ChemRLformer (Ghugare et al., 2023) and Taiga (Mazuz et al., 2023) utilize RL to fine-tune an NN that was pre-trained to generate SMILES strings. RL guides the generation towards molecules with a high value of the property of interest (e.g., QED, DS). MolDQN (Zhou et al., 2019) belongs to the family of sequential generation methods based on RL. It uses a modification of the Deep Q-Learning algorithm (Mnih et al., 2013) to sequentially select atoms and bonds to add to or remove from the molecular graph. A new promising alternative to RL-based generation are Generative Flow Networks (Bengio et al., 2021; Jain et al., 2023). These models are trained to sample objects with probabilities proportional to the reward function. Unlike RL-based methods, Generative Flow Networks are capable of sampling from different modes of the distribution which is especially useful in drug design applications. Pocket2Mol (Peng et al., 2022) is trained in an autoregressive fashion to predict masked atoms of drug molecules in protein-ligand pairs. The trained model sequentially generates the atom's position, type, and bonds. The generation process is conditioned on both the protein target and the previously generated atoms. These methods have demonstrated their ability to generate potential drug candidates that selectively target specific proteins. We compare FREED++ with all the approaches described in this section except DiffSBDD, which has been shown (Schneuing et al., 2023) to produce molecules inferior to baselines in terms of various metrics.

## 3 FREED++

This section provides a detailed description of the proposed FREED++ method. We highlight the differences between the original FREED and the proposed FREED++ with blue boxes. In section 4, we explain the bugs in the original implementation of the FREED model and discuss their effects on the model's performance.

---

[1] https://github.com/AIRI-Institute/FFREED

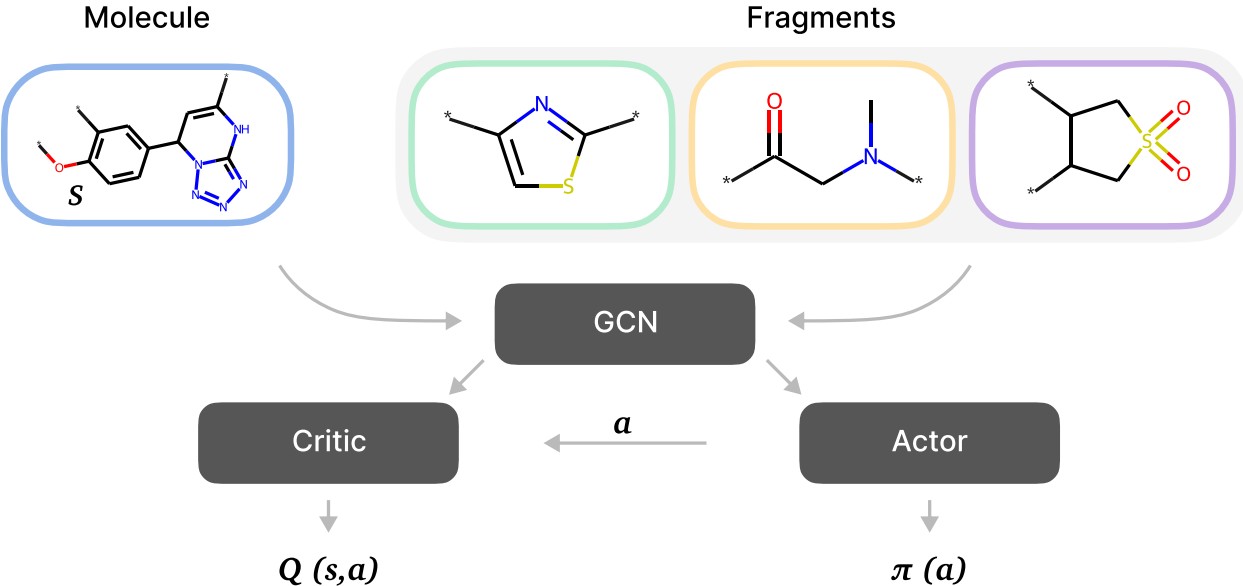

Figure 2: Overview of fragment-based molecule generation frameworks. At each step, a fragment is selected from a pre-defined fragment library $\mathcal{F}$ and attached to the current state $s$. In general, different encoding methods may be used to handle $s$ and the selected fragment $f$, but in this work, both are processed with the same GCN.

FREED++ is a fragment-based molecule generation framework; its overview is shown in Fig. 2. It operates on fragments instead of single atoms to speed up the generation process and ensure the validity of generated molecules. The differences between FREED++ and the original FREED are described in sections 4.1,4.2, 4.3.

## 3.1   MDP

The state space $\mathcal{S}$ is defined as a set of all possible extended molecular graphs describing stable molecules (see section 3.2). The action $a$ consists of selecting a fragment $f$ and concatenating it with the molecular graph $s$. The action comprises three steps: "selecting where to attach a new fragment $(a_1)$", "choosing which fragment to attach $(a_2)$", and "determining where on a new fragment to form a chemical bond $(a_3)$". More details about the action can be found in section 3.3. The initial state $s_0$ is a benzene ring with 3 attachment points. Transition dynamics is straightforward: given a graph representation $s_t$ of the current state of the molecule and an action $a_t = (a_t^1, a_t^2, a_t^3)$, the new graph is assembled from $s$ and the new fragment via RDKit (Landrum et al., 2022). The length of the episodes is fixed and equals $T$. In the original implementation, the same length $T = 4$ of episodes is used for all proteins (see section J for discussion). The reward $r_t$ is calculated as follows:

$$r_t(s_t, a_t, s_{t+1}) = \begin{cases} 0, \text{ if } t < T; \\ \max(0, DS(s_{t+1})), \text{ if } t = T. \end{cases} \quad (1)$$

We define the DS as the negative binding energy estimated by the docking software. $DS(s)$ is calculated in three steps. First, initial 3D conformation for docking is generated for a given molecular graph with OpenBabel (O'Boyle et al., 2011). Then, the binding affinity of the molecule to the target protein is estimated with QuickVina2 (Alhossary et al., 2015). Details on reward computation are in Appendix H. Lastly, we take the negative value of the estimated binding affinity to get the DS.

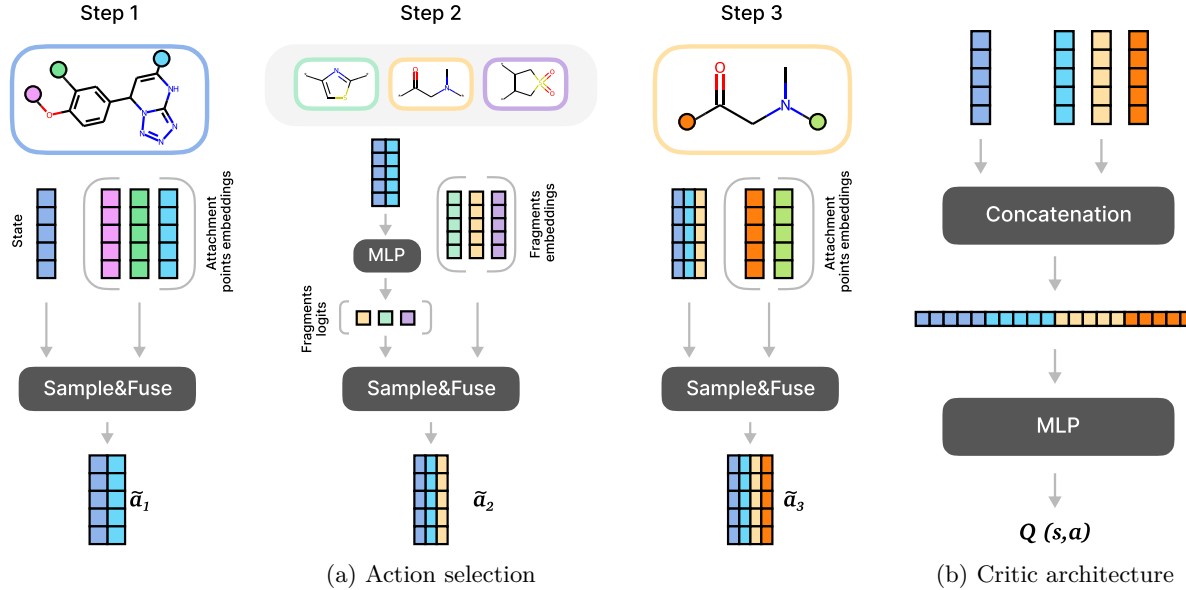

(a) Action selection          (b) Critic architecture

Figure 3: A schematic overview of the actor and the critic. The action selection process is depicted in Fig. 3a. **Step 1**: the current state $s$ is embedded with a GCN $G_\theta$, and the resulting embedding is fused with the embeddings of its attachment points. Then, one of the attachment points is selected as $a_1$, and its embedding $\tilde{a}_1$ is used in the next step. **Step 2**: $\tilde{a}_1$ is passed through an MLP to get a distribution over the available fragments. One of the fragments is selected as $a_2$, and its embedding $\tilde{a}_2$ is used in the next step. **Step 3**: the selected fragment $a_2$ is processed by the same GCN $G_\theta$ to obtain the embeddings of its attachment points. After fusing these embeddings with $\tilde{a}_2$, one of the attachment points of the fragment is selected as $a_3$. **Critic**: The embeddings of all actions are concatenated with the state embedding and processed by the critic (Fig. 3b).

## 3.2 State

FREED operates on extended molecular graphs — graphs that contain a special type of node called an "attachment point". These points indicate the locations where the atomic bonds were severed during the fragmentation process. Consequently, the state is represented as a molecular graph with attachment points. When the generation process is completed, all attachment points (if present) are replaced with hydrogen atoms.

To be processed by a Graph Convolution Network (GCN) , a graph needs to be converted to a tensor. This is achieved by representing the nodes and edges with a set of feature vectors and an adjacency matrix. The node features encompass one-hot encodings of the atom's type, valency, degree, the count of hydrogen atoms, and a flag indicating whether the atom is a part of an aromatic ring. Edge features consist of one-hot encoding of the bond type.

## 3.3 Action

The action $a = (a_1, a_2, a_3)$ is a composite of three elements. The first component $a_1$ is the index of an attachment point to which the new fragment is attached. To obtain a categorical distribution over the attachment points of state $s$, we first embed $s$ with a GCN (Kipf & Welling, 2017): $\tilde{V} = G_\theta(s)$. We then use the resulting matrix of node embeddings $\tilde{V} \in \mathbb{R}^{|V| \times d}$ in two ways: 1) an aggregated graph embedding $\tilde{S} = \mathbf{agg}(\tilde{V}), \tilde{S} \in \mathbb{R}^d$ is derived from $\tilde{V}$, where **agg** is an aggregating function (normally, a sum or an average of rows); 2) a submatrix $\tilde{V}^{\mathrm{att}} \in \mathbb{R}^{|V^{\mathrm{att}}| \times d}$ of node embeddings corresponding to attachment point is formed. $|V|$ and $|V^{\mathrm{att}}|$ are the total number of nodes in the graph and the number of attachment points, respectively, and $d$ is the embedding size. $\tilde{S}$ and $\tilde{V}^{\mathrm{att}}$ are then combined via a trainable fusing (i.e., Multiplicative interactions (Jayakumar et al., 2020) layer or a concatenation layer; refer to section 4.3 for an in-depth

discussion) function $\mathbf{f}_{\phi_1}$, resulting in $\tilde{A}_1(s) \in \mathbb{R}^{|V^{\text{att}}| \times d}$:

$$\tilde{A}_1(s) = \mathbf{f}_{\phi_1}(\tilde{S}, \tilde{V}^{\text{att}}). \tag{2}$$

After that, we use an MLP $\mathbf{f}_{\psi_1} : \mathbb{R}^{|V^{\text{att}}| \times d} \to \mathbb{R}^{|V^{\text{att}}|}$ that maps $\tilde{A}_1(s)$ to logits that define a categorical distribution over attachment points. The Gumbel-Softmax (Jang et al., 2017) operator $\sigma$ is used to reparameterize the resulting distribution. Categorical reparameterization is essential because $a_2$ and $a_3$ are sampled autoregressively and depend on previously sampled discrete actions.

$$p_1(\cdot \mid s) = \sigma(\mathbf{f}_{\psi_1}(\tilde{A}_1(s))). \tag{3}$$

The second component $a_2$ is the index of the fragment to be attached to $s$. First, we sample action $a_1$ from $p_1(\cdot \mid s)$ (Eq. 3) and select the $a_1$-st row $\tilde{a}_1$ of the matrix $\tilde{A}_1(s)$. To obtain the logits of the categorical distribution over the fragment candidates, we process $\tilde{a}_1$ with an MLP $\mathbf{f}_{\psi_2} : \mathbb{R}^d \to \mathbb{R}^{|\mathcal{F}|}$, where $\mathcal{F}$ is a set of all available fragments, and $|\mathcal{F}|$ is their total number.:

$$p_2(\cdot \mid s, a_1) = \sigma(\mathbf{f}_{\psi_2}(\tilde{a}_1)). \tag{4}$$

> In FREED, the categorical distribution over fragment candidates is computed similarly to the distribution over attachment points (see Section 4.3). First, ECFP embeddings (Rogers & Hahn, 2010) $\tilde{F}^{\text{ECFP}}$ are calculated for all the fragments. Then, $\tilde{F}^{\text{ECFP}}$ and $\tilde{a}_1$ are combined via fusing funtion:
>
> $$\tilde{A}_2 = \mathbf{f}_{\phi_2}(\tilde{a}_1, \tilde{F}^{\text{ECFP}}). \tag{5}$$
>
> After that, an MLP $\mathbf{f}_{\psi_2} : \mathbb{R}^{|\mathcal{F}| \times d} \to \mathbb{R}^{|\mathcal{F}|}$ maps $\tilde{A}_2(\tilde{a}_1)$ to logits which define a categorical distribution over fragments.
>
> $$p_2(\cdot \mid s, a_1) = \sigma(\mathbf{f}_{\psi_2}(\tilde{A}_2)). \tag{6}$$

Note that if BRICS fragmentation (see section 5.2) is used, logits associated with fragments that lead to invalid molecules are replaced with $-\infty$, effectively preventing the sampling of such fragments. Then, we sample an index $a_2 \sim p_2(\cdot \mid s, a_1)$ and the corresponding fragment $f_{a_2}$ and process it along with all available fragments (to reuse in case a batch of fragments is sampled) with the same GCN $G_\theta$ to get matrix $\tilde{F} \in \mathbb{R}^{|\mathcal{F}| \times d}$ of aggregated embeddings:

$$\tilde{F} = \{\mathbf{agg}(G_\theta(f))\}_{f \in \mathcal{F}}, \tag{7}$$

Then, the $a_2$-nd row of matrix $\tilde{F}$ is selected and combined with $\tilde{a}_1$ using a fusing function $\mathbf{f}_{\phi_2}$:

$$\tilde{a}_2(s, a_1) = \mathbf{f}_{\phi_2}(\tilde{a}_1, \tilde{F}_{a_2}). \tag{8}$$

> Instead of fusing $\tilde{a}_1$ with the GCN embedding of the selected fragment $a_2$, the original implementations uses a fixed ECFP emdding:
>
> $$\tilde{a}_2(s, a_1) = \mathbf{f}_{\phi_2}(\tilde{a}_1, \tilde{F}_{a_2}^{ECFP}). \tag{9}$$

Unlike in "Step 1", where we first fuse the state and embeddings of attachment points and then sample $a_1$, in "Step 2", we first sample $a_2$ and then fuse its embedding $\tilde{F}_{a_2}$ with $\tilde{a}_1$. This simplification (see section 4.3) allows to speed up the training process significantly.

The third component $a_3$ is the index of an attachment point on the selected fragment $f_{a_2}$. To obtain a categorical distribution over attachment points of the selected fragment $f_{a_2}$, we first retrieve the matrix of node embeddings of the selected fragment $\tilde{F}_{a_2} = G_\theta(f_{a_2})$. We then select node embeddings corresponding to attachment points $\tilde{F}_{a_2}^{\mathrm{att}} \in \mathbb{R}^{|f_{a_2}^{\mathrm{att}}| \times d}$, where $|f_{a_2}^{\mathrm{att}}|$ is the number of attachment points on the selected fragment $f_{a_2}$. A fusing function $\mathbf{f}_{\phi_3}$ is utilized to combine this embedding with $\tilde{a}_2$:

$$\tilde{A}_3(s, a_1, a_2) = \mathbf{f}_{\phi_3}(\tilde{a}_2, \tilde{F}_{a_2}^{\mathrm{att}}). \tag{10}$$

After that, we employ an MLP $\mathbf{f}_{\psi_3} : \mathbb{R}^{|f_{a_2}^{\mathrm{att}}| \times d} \to \mathbb{R}^{|f_{a_2}^{\mathrm{att}}|}$ that maps $\tilde{A}_3(s, a_1, a_2)$ to logits that define a categorical distribution over the attachment points of the selected fragment. Similarly to "Step 2", if BRICS fragmentation is used, logits corresponding to invalid attachment points are replaced with $-\infty$.

$$p_3(\cdot \mid s, a_1, a_2) = \mathbf{f}_{\psi_3}(\tilde{A}_3(s, a_1, a_2)). \tag{11}$$

Lastly, we sample $a_3 \sim p_3(\cdot \mid s, a_1, a_2)$ and select the $a_3$-rd row $\tilde{a}_3$ of the matrix $\tilde{A}_3(s, a_1, a_2)$. The whole action selection process is illustrated in Fig. 3a.

### 3.4 Critic

The critic architecture is illustrated in Fig. 3b. It consists of two components: an encoder GCN $G_\theta$, which is also used for action selection, and an MLP $\mathbf{f}_\omega : \mathbb{R}^{4d} \to \mathbb{R}$. To ensure training stability, the encoder $G_\theta$ is frozen in the actor and is only trained as a part of the critic. In FREED, the MLP $\mathbf{f}_\omega$ takes the concatenation of $(\tilde{S}, \tilde{V}_{a_1}^{\mathrm{att}}, \tilde{F}_{a_2}, (\tilde{F}_{a_2}^{\mathrm{att}})_{a_3})$ and outputs the $Q$-value.

There are two scenarios in which the critic is used. The first scenario involves applying the critic to the actions saved in the replay buffer. Since the saved action $(a_1, a_2, a_3)$ is just a set of indices, the critic needs to reprocess the saved state and fragment with the current version of the GCN and then pass it to $\mathbf{f}_\omega$. This is necessary because the GCN parameters $\theta$ may have been updated since the transition was saved in the replay buffer. The second scenario involves applying the critic to actions generated with up-to-date $G_\theta$ (i.e., when calculating actor loss in SAC). In this case, there is no need for reprocessing, and the concatenation of $(\tilde{S}, \tilde{V}_{a_1}^{\mathrm{att}}, \tilde{F}_{a_2}, (\tilde{F}_{a_2}^{\mathrm{att}})_{a_3})$ is passed directly to $\mathbf{f}_\omega$.

> In the original FREED paper, crtitic architecture was not covered. For the description of the critic used in FREED refer to the section 4.1.

## 4 Fixing FREED

This section summarizes all the changes made to the original FREED model. This section is divided into three subsections. In the first subsection, we describe the issues and bugs found in the original implementation of FREED, along with their potential effects on the model's performance. In the second subsection, we describe minor changes introduced to simplify the model, reduce the number of hyperparameters, or improve the the training stability. By implementing all the changes described in 4.1, 4.2, we arrive at a fixed version of FREED, which we dub FFREED".

In 4.3, we describe various components of FFREED that can be removed or simplified to speed up the model and reduce the number of trainable parameters. The resulting model after all the simplifications is called "FREED++".

### 4.1 Major implementation issues

**Critic architecture** In the original implementation, the critic is parameterized as an MLP, which takes as input a concatenation of four vectors: 1) an embedding of the molecule generated by $G_\theta$; 2) probabilities

associated with attachment points on $s$ (see eq.3); 3) one-hot encoding of the selected fragment $f_{a_2}$; and 4) probabilities associated with attachment points on $f_{a_2}$ (see eq. 11).

First, the critic receives no information about the selected attachment points, as it operates on probabilities. Furthermore, the order in which the probabilities for the attachment points are passed to the critic is determined by the inner representation of the current state $s$ in RDKit (Landrum et al., 2022). This order may change as new fragments are added throughout the trajectory. Due to these two factors, the critic cannot attribute high rewards to selecting specific attachment points. Consequently, learning any meaningful policy for selecting $a_1$ and $a_3$ becomes impossible. Refer to 3.4 for details of our proposed implementation of the critic architecture.

**Critic update**   The agent is trained with the SAC (Haarnoja et al., 2018a) algorithm that operates inside the Maximum Entropy Framework (Ziebart et al., 2008; Haarnoja et al., 2017), which augments the standard maximum reward reinforcement learning objective with an entropy maximization term. The target for the $Q^\pi$ function includes terms responsible for both the future discounted reward and the entropy of the policy:

$$\hat{Q}^\pi(s_t, a_t) = \mathbb{E}_{(s_t, a_t, s_{t+1}) \sim \mathbb{D}, \tilde{a} \sim \pi(\cdot | s_{t+1})} \left[ r(s_t, a_t, s_{t+1}) + \gamma \left( Q^\pi(s_{t+1}, \tilde{a}) - \alpha \log \pi(\tilde{a} \mid s_{t+1}) \right) \right]. \tag{12}$$

In the original implementation, the entropy term is omitted from the equation. This way, at the policy improvement step, the actor is forced to maximize the cumulative return and ignore the term responsible for the entropy of the policy in the succeeding states.

**Target networks usage**   Target networks are used to stabilize the training of $Q$ functions (Mnih et al., 2015). A common practice for the SAC algorithm to update the target network throughout training smoothly: $Q^\pi_{\text{targ}} = (1 - \tau) Q^\pi_{\text{targ}} + \tau Q^\pi$; where $\tau$ is a smoothing constant that is usually set to a small value between 0.01 and 0.001 (Haarnoja et al., 2018a). Instead, in the original implementation, $\tau$ is set to 1, which effectively means that no target network was used. This leads to the moving target problem and destabilizes the training.

**Gumbel-Softmax**   Recall (see section 3.3) that the action proposed in FREED consists of three discrete random variables which are sequentially sampled in an autoregressive fashion. The authors employed Gumbel-Softmax (Jang et al., 2017) to propagate the gradients through the discrete sampling process.

We found two issues in the original implementation. First, probabilities were used instead of logits. Second, a parameter $\nu$ was introduced without a clear motivation. We show that sampling from the Gumbel-Softmax distribution with these two issues is equivalent to sampling $y \propto \exp \frac{\pi}{\nu}$ with a temperature $\frac{\tau}{\nu}$, where $\pi$ is the desired discrete distribution, and we use $\propto$ as $\exp \frac{\pi}{\nu}$ does not necessarily define a distribution. Refer to Appendix B for the derivation and an in-depth discussion of the effects.

To evaluate the significance of each particular major implementation issue, we perform an ablation study in Appedinx K.

### 4.2   Minor issues and simplifications

**Message passing**   As states and fragments represent extended molecular graphs, we use a GCN to obtain embeddings. GCN iteratively updates node representations by aggregating the information from the node's local neighborhood. In the original implementation, directed graphs are used instead of undirected ones. The direction of the edge is determined by the inner representation of the graph in RDKit (Landrum et al., 2022). Such molecular representation differs from the undirected one previously used in various applications of ML and DL in CADD (Sun et al., 2020; Wieder et al., 2020). The intuition behind the usage of undirected molecular graphs lies in the fact that chemical bonds do not have naturally imposed directions. Moreover, Wu et al. (2018) have previously shown that treating molecules as undirected graphs improves performance in predicting various molecular properties. We use undirected molecular graphs.

**Learning rate schedulers**    In the original implementation, the learning rate scheduler is used. The actor learning rate is decreased by a factor of 10 if the training actor loss has not improved for a certain number of steps. This is not an optimal choice for actor-critic RL algorithms, as actor loss is poorly correlated with agent performance. While studying the original implementation, we noticed that for some target proteins, such a scheduler caused the learning rate to converge quickly at the beginning of training, preventing the agent from improving further. We use a constant learning rate throughout the training to mitigate this issue and simplify the selection of hyperparameters.

**Reward shaping**    In the original implementation, reward shaping is used. While investigating the proposed reward shaping approach, we found it to be meaningless (see A.7) and removed it. Instead, we only assign non-zero rewards to the terminal states (see 3.1).

**Temperature hyperparameter in SAC**    In the original implementation, the temperature $\alpha$ is only trained for a fraction of the whole training time, and, furthermore, $\alpha$ is clipped to be in the range $[0.05, 20]$ (details in A.5). We follow the original implementation, remove clipping, and train $\alpha$ throughout training.

**Architecture**    In the original implementation, the sizes of latent spaces are decreased quickly and deeper network for fragment selection is used compared to the selection of attachment points. We operate on bigger embeddings and unify the structure of logits projectors for all actions. The exact architecture changes and their effects are discussed in section A.9).

Other minor changes and simplifications can be found in the Appendix A.

### 4.3    From fixed FREED to FREED++

In this section, we describe how to significantly speed up the FFREED model and reduce the number of trainable parameters. We investigate several components of the original FREED framework and show that they can be removed or simplified without a performance drop while significantly speeding up the model and reducing the number of trainable parameters.

**Prioritization**    To encourage the agent to generate diverse molecules, the original paper's authors propose their variant of prioritized experience replay (Schaul et al., 2015). Instead of using the TD-error to determine the probability of sampling a transition from the replay buffer, the authors suggest prioritizing novel transitions. The novelty in terminal states is defined as the $L_2$ error between the reward in the terminal state and the predicted reward. In non-terminal states, the non-existent reward is approximated with the $Q$ function, and the $L_2$ error between its output and the predicted reward is used as a novelty. Apart from introducing additional computational load, this approach has multiple issues, which we discuss in Appendix C. We train the agent without PER and sample transitions uniformly.

**Fusing functions**    As mentioned in 3.3, fusing functions $\mathbf{f}_{\phi_1}$, $\mathbf{f}_{\phi_2}$, and $\mathbf{f}_{\phi_3}$ are used when selecting $a_1$ and $a_3$ to combine the embedding of a state or a fragment with the embedding of an attachment point (see equations 2, 10). In the original paper, multiplicative interactions (Jayakumar et al., 2020) are used as fusing functions. Let $x \in \mathbb{R}^{d_1}, z \in \mathbb{R}^{d_2}$ represent the embeddings to be combined. Then, the MI of $x$ and $z$ is:

$$\mathbf{f}_{\phi_i}^{MI}(x, z) = z^T \mathbf{W}_i x + \mathbf{U}_i z + \mathbf{V}_i x + \mathbf{b}_i, \tag{13}$$

where $\mathbf{W} \in \mathbb{R}^{d_2 \times d_3 \times d_1}$ is a learnable 3D tensor, $\mathbf{U} \in \mathbb{R}^{d_3 \times d_2}, \mathbf{V} \in \mathbb{R}^{d_3 \times d_1}$ are learnable matrices, and $\mathbf{b} \in \mathbb{R}^{d_3}$ is a bias term. Note that if not stated otherwise, $d_1 = d_2 = d_3 = d$. Notice that the MI layer contains a bilinear form, which is considered to be computationally and memory expensive. To overcome this drawback, we simplify the fusing function and replace the MI layer with a concatenation layer:

$$\mathbf{f}_{\phi_i}^{CAT}(x, z) = \mathbf{Q}_i [x; z] + \mathbf{b}_i, \tag{14}$$

where $[\cdot \, ; \cdot]$ denotes the concatenation operator, $\mathbf{Q}_i \in \mathbb{R}^{d_3 \times (d_1 + d_2)}$ is a learnable matrix, and $\mathbf{b} \in \mathbb{R}^{d_3}$ is a bias term.

**Fragment selection** In the original implementation of FREED, the fragment selection protocol resembles the procedure of attachment point selection. First, the ECFP representations of the fragments $\tilde{F}^{\mathrm{ECFP}} = \{\mathrm{ECFP}(f)\}_{f \in \mathcal{F}}$ are built. ECFP is a function that computes a $d_2 = 1024$-bit Morgan fingerprint of radius 2 (Morgan, 1965). Then, the embeddings of all fragments are fused with $\tilde{a}_1$: $\tilde{A}_2 = \mathbf{f}_{\phi_2}(\tilde{a}_1, \tilde{F}^{\mathrm{ECFP}})$, and the resulting matrix $\tilde{A}_2$ is processed with an MLP $\mathbf{f}_{\psi_2} : \mathbb{R}^d \to \mathbb{R}$ to obtain the logits of the distribution on the fragments. Notice that when $d_2$ is large, the bilinear form in equation 13 becomes expensive to compute. To avoid that, we replace the procedure described in the original paper with the one described in equation 4.

In summary, by applying these three changes to FFREED, we get FREED++. We carefully compare FREED++ with different variants of FFREED in section F.

## 5 Experiments

In this section, we describe the experiments conducted and our evaluation protocol. We thoroughly tested the improved FREED++ framework on various tasks related to generating molecules with desired properties. As binding affinity is a key property of a molecule that defines the therapeutic effect of a drug, we consider docking score optimization as the main objective in our experiments.

**Comparison with baselines** First, we compare FREED++ with existing molecular generative baselines on the task of generating molecules with high affinity against particular biological targets 5.1. We consider this to be the main experiment.

**Fragment library collection** Existing tools for *de novo* drug design use various libraries of building blocks with sizes ranging from dozens to several thousands of fragments (Rotstein & Murcko, 1993; Pierce et al., 2004; Douguet et al., 2005; Spiegel & Durrant, 2020; Yuan et al., 2020). The main reasons behind the choice of fragment library can be summarized as follows: 1) the building block library should be reasonably small to reduce the chemical space efficiently, and 2) if known inhibitors for a particular target are available, it is better to use their fragments for assembling. (Lin, 2000). While the used fragment library essentially defines the set of attainable molecules, research papers on *de novo* drug design focus mainly on scoring functions, search algorithms, and assembling strategies (Schneider & Fechner, 2005). To improve the understanding of the problem, we conducted experiments with several fragment libraries obtained from two molecular datasets with different fragmentation techniques 5.2.

**Development of USP7 inhibitors** Finally, we have tested FREED++ in the practical scenario of generating inhibitors for the USP7 protein and provided qualitative analysis of the generated molecules 5.3.

**Protein targets** In the original paper, three protein targets are considered: fa7 (Zbinden et al., 2005), parp1 (Penning et al., 2010), and 5ht1b (Wang et al., 2013). To show an improved generalization, we experiment with three additional proteins: abl1 (Cowan-Jacob et al., 2007) and fkb1a (Sun et al., 2003) from DUDE (Mysinger et al., 2012) and usp7 (Leger et al., 2020a) from PDB (Berman et al., 2000). Binding pockets of interest for proteins were computed as follows: first, we extracted the 3D structure of the ligand from the corresponding pdb file; then, we computed the center of the bounding box as the average of all atoms' coordinates. The size of the bounding box along each axis is estimated by adding the maximum difference between the coordinates of the atoms and 4Å for the corresponding axis.

**Evaluation protocol** To evaluate the generation quality, we generate 1000 molecules with a fully-trained model, remove invalid compounds and duplicates and score the filtered molecules with QVina 2. We repeat the generation 3 times with different random seeds. Since the combinatorial generator is a simple baseline, we allow it to generate 30000 molecules instead of 1000.

It is important to note that the evaluation protocol differs from the one used in the original paper. In the original work, all models except FREED are compared on the first 3000 molecules generated during the training. The FREED is evaluated on the 3000 molecules generated after the exploration phase (see section A.3). Such an approach does not reflect the actual performance of models, as the evaluated models are

underfitted. For example, the default REINVENT model generates $\sim 1.8 \times 10^5$ molecules throughout the training, requiring approximately twice as much computational time as FREED.

**Metrics** To evaluate the performance of different models, we report the uniqueness of valid molecules, the average docking score of unique and valid molecules (Avg DS), the maximum docking score (Max DS) and the average docking score of 5% of top-scoring molecules (Top-5 DS). We report DS metrics in KCal/Mol units. In section G, we also consider PAINS, SureChEMBL and Glaxo metrics, which denote the fraction of valid unique molecules that successfully pass the corresponding structural filters.

### 5.1 Comparison with baselines

In this section, we compare the model corresponding to the original implementation[2], which we call FREED 4, fixed model FFREED 4.3, and our simplified model FREED++ 3 on the task of generating high-affinity molecules.

**Baselines** We take the same objective-oriented baselines as in the original paper: REINVENT and MolDQN. Additionally, we consider two methods: the combinatorial generator (random walk), which selects fragments proportional to their frequences at each step, and one of the current SOTA protein-conditioned generative methods Pocket2Mol (Peng et al., 2022). Moreover, we report metric values computed over sets of known inhibitors (KI rows in tables 1, 6). For fa7, parp1, 5ht1b, abl1, and fkb1a targets we take inhibitors from the DUDE dataset, and for the usp7 protein we take inhibitors reported in the paper (Leger et al., 2020b).

For the combinatorial generator (CombGen) we used the implementation from MOSES[3]. For Pocket2Mol we used a pretrained model provided by the authors[4].

We train REINVENT with DS as the optimization objective. For non-valid molecules, DS is considered to be 0. We train REINVENT with the default parameters of the original implementation[5] except for the batch size (we take 32 instead of 64). As the reward we take $0.1 \cdot DS(s_t)$ for valid molecules and 0 otherwise, to keep a return approximately in the range [0, 1]. We search for the optimal $\sigma \in [60, 100, 200]$ (see table 5) and pick the best in the evaluation.

Regarding MolDQN, we explore two setups. The first setup is similar to the original implementation[6], except we replace QED with DS. In this version, gradient steps are performed every 20 iterations, and the reward at each step is calculated as $\gamma^{T-t}DS(s_{t+1})$. The second version employs sparse rewards, with rewards set to 0 everywhere except for preterminal states, where the reward is $DS(s_{t+1})$. In the "sparse" setup, we do 24 gradient steps every 480 iterations maintaining the same update-to-data ratio. From the training plots (see 5), we observe that the version from the original implementation performs better but is approximately three times slower. We use the "sparse" version setup in the Docking score optimization section. Overall, changing the "sparse" version to the dense one does not alter the results.

**Results** The results of the experiment are presented in tables 1, 6. To sum it up, the corrections to the FREED model (FFREED and FREED++) perform superior to other methods in all metrics except uniqueness. However, we agree with (Peng et al., 2022) that uniqueness and diversity are not very important metrics for the pocket-based generation task because protein pocket is known to have strong specificity. Moreover, FREED++ allows controlling the trade-off between diversity and generation quality via the target entropy parameter.

While Pocket2Mol is trained to recover the masked atoms of the known inhibitors, objective-oriented methods REINVENT, MolDQN, FFREED, and FREED++ are directly aimed to optimize the docking score through the design of the reward function. Although known inhibitors commonly form strong interactions with their

---

[2]https://github.com/AITRICS/FREED
[3]https://github.com/molecularsets/moses
[4]https://github.com/pengxingang/Pocket2Mol
[5]https://github.com/MarcusOlivecrona/REINVENT
[6]https://github.com/aksub99/MolDQN-pytorch

target proteins, there are no guarantees that they have the highest possible docking scores, which we also observe in our experiments. Therefore, one can expect that RL-based methods will be more successful in the DS optimization task. We partially observed such an effect in our experiments, except MolDQN, which failed to generate molecules with higher docking scores than Pocket2Mol. Possible reasons for this could be the underfitting of models, short episode length, poor choice of hyperparameters, or the use of a non-pre-trained encoder. While REINVENT succeeded in outperforming Pocket2Mol, it uses a backbone pre-trained on a large database compared to MolDQN, which learns from scratch.

We observe that FREED++ and FFREED outperform MolDQN and REINVENT. We hypothesize that this behavior can be explained by the fact that FREED methods work in the paradigm of fragment generation. Fragment generation exponentially reduces the search space compared to atom-based assembling strategies, allowing the RL agent to learn efficiently.

We found that the original FREED model performs significantly worse than other models. The learning process of FREED tends to diverge (see figure 6), while FFREED and FREED++ stably outperform their competitors. This confirms the importance of correcting implementation issues (sec. 4). We compared FREED and FREED++ on the first 10 proteins from the DUDE database to provide more evidence (see figure 7). Once again, FREED poorly optimizes the docking score and works unstably.

| method | unique ($\uparrow$) | Avg DS ($\uparrow$) | Max DS ($\uparrow$) | Top-5 DS ($\uparrow$) |
|---|---|---|---|---|
| CombGen | $0.98 \pm 0.00$ | $8.03 \pm 0.00$ | $13.00 \pm 0.26$ | $10.53 \pm 0.00$ |
| MolDQN | $0.21 \pm 0.03$ | $7.92 \pm 0.23$ | $9.93 \pm 0.41$ | $9.43 \pm 0.34$ |
| REINVENT | $0.99 \pm 0.00$ | $9.69 \pm 0.39$ | $13.13 \pm 0.89$ | $12.05 \pm 0.73$ |
| Pocket2Mol | $1.00 \pm 0.00$ | $8.71 \pm 0.37$ | $12.36 \pm 0.40$ | $11.30 \pm 0.65$ |
| FREED | $0.10 \pm 0.07$ | $8.58 \pm 1.02$ | $11.36 \pm 0.98$ | $11.12 \pm 0.88$ |
| FFREED | $0.66 \pm 0.01$ | $\mathbf{10.91 \pm 0.15}$ | $14.03 \pm 0.46$ | $\mathbf{13.35 \pm 0.27}$ |
| FREED++ | $0.64 \pm 0.07$ | $9.66 \pm 3.28$ | $\mathbf{14.16 \pm 0.47}$ | $13.30 \pm 0.13$ |
| KI | - | $9.40$ | $11.90$ | $11.90$ |

Table 1: Comparison of FREED, FFREED and FREED++ with baselines on DS optimization task for USP7 target. See full table in appendix 6.

## 5.2 Fragment library collection

An appropriate fragment library is an essential part of a successful molecular generation. Such a set of fragments can be handcrafted by a chemist or obtained via a fragmentation procedure applied to a given set of molecules.

To obtain fragments for generation, the authors of FREED use CReM fragmentation (Polishchuk, 2020) to fragment 250k drug-like molecules from the ZINC (Irwin et al., 2012) database. Additionally, the authors filter fragments according to the number of atoms, radius, and frequency. Fragments that contain fewer than 12 atoms or appear less than 3 times in the ZINC are excluded. Fragments that cause RDKit parsing errors are also removed. When two fragments have the same graph structure but different attachment sites (almost duplicates), the one with fewer attachment sites is removed. Finally, the authors choose 91 fragments that appear most frequently from the filtered dataset.

CReM fragmentation is not widely used, and formally, the given dataset cannot be reconstructed with fragments obtained by CReM. BRICS fragmentation (Degen et al., 2008) is a set of rules for breaking retrosynthetically interesting chemical substructures commonly used for *de novo* design.

We fragment MOSES ( (Polykovskiy et al., 2020) - cleaned version of ZINC) and ZINC datasets by CReM and BRICS fragmentation procedures and compare different combinations of fragmentation and dataset. The same fragment dictionary as in the previous section was used for CReM-ZINC setup. For other setups, we do

the following steps: extract fragments from the dataset, exclude charged fragments, remove fragments that appear only once, remove fragments with more than 16 heavy atoms, remove almost duplicate fragments, and sample 100 fragments uniformly.

In FREED++, the optimal number of generation steps is unknown beforehand and needs to be tuned (see section J for an in depth discussion). Because of that, we run the generation for 4 and 5 steps and use the one which produces molecules with higher docking scores on evaluation.

| fragmentation | dataset | unique (↑) | Avg DS (↑) | Max DS (↑) | Top-5 DS (↑) |
|---|---|---|---|---|---|
| BRICS | MOSES | $0.58 \pm 0.17$ | $5.82 \pm 4.32$ | $13.16 \pm 0.51$ | $12.22 \pm 0.67$ |
| | ZINC | $0.63 \pm 0.13$ | $9.07 \pm 0.70$ | $13.39 \pm 0.88$ | $12.42 \pm 0.46$ |
| CReM | MOSES | $0.74 \pm 0.07$ | $1.21 \pm 1.06$ | $12.00 \pm 1.81$ | $7.77 \pm 6.27$ |
| | ZINC | $0.64 \pm 0.07$ | $\mathbf{9.66 \pm 3.28}$ | $\mathbf{14.16 \pm 0.47}$ | $\mathbf{13.30 \pm 0.13}$ |

Table 2: Comparison of different fragments libraries used in FREED++ on DS optimization task for USP7 target. See full table in appendix 7.

**Results**   Tables 2 and 7 demonstrate that generation performance is significantly influenced by the fragment library used. We explain these results by drawing an analogy between molecular fragmentation and tokenization in the NLP field. Both procedures break unstructured data into a set of meaningful elements, which are used as atomic units that embed contextual information. It is known that tokenization has a major impact on overall pipeline performance in NLP tasks  (Chirkova & Troshin, 2023; Zhang & Li, 2021; Tay et al., 2022; Park et al., 2020). Based on our empirical results, we conclude that the design of a fragment library is an extremely significant step in molecular generation.

### 5.3   Development of USP7 inhibitors

USP7 is a deubiquitinating enzyme (DUB) which takes part in regulating of a plethora of biological processes. The up-regulated function of the protein is related to the development of several types of cancer, thus, inhibition of USP7 is considered a promising strategy for the treatment of these malignancies. During the last decade, numerous series of USP7 inhibitors were developed, however, none of them progressed into clinical trials due to their insufficient efficacy and selectivity. For that reason, it is still highly desired to develop novel structural types of USP7 inhibitors which possess improved profile of pharmacodynamic and pharmacokinetic properties. Known USP7 (see figure 9) inhibitors bind (Li & Liu, 2020; Oliveira et al., 2022; Korenev et al., 2022) to the protein in three different locations: inside the catalytic center (some irreversible inhibitors form a covalent bond with sulfur atom for catalytic Cys223); in the ubiquitin-binding cleft near the catalytic site and in the allosteric binding site (4-ethylpyridine series of inhibitors).

A highly promising structural type of USP7 inhibitors was reported in 2020 (Leger et al., 2020b). Although these molecules bind to the familiar USP7 pocket, the outstanding selectivity relative to other DUBs was demonstrated together with notable efficacy *in vivo*. Those inhibitors fit the targeted pocket well and form four well-defined hydrogen bonds. In this work, we utilize FREED++ to generate novel plausible inhibitors of USP7 which bind to the same binding site.

We perform the generation procedure for several fragment libraries: with the basic set of fragments (CReM-ZINC), with a relatively big set of fragments (1000 fragments) from the MOSES dataset (BRICS-MOSES) and the set of fragments obtained via fragmentation of known inhibitors of usp7 (BRICS-USP7). We construct the reward function in a way that takes into account several commonly desired properties of drug candidates. We search for molecules 1) with octanol-water partition coefficient LogP $\in [0, 5]$; 2) with a number of heavy atoms $\leq 40$ 3) with a number of hydrogen acceptors $\leq 10$; 4) a number of hydrogen donors $\leq 5$; 5) which pass PAINS, SureChEMBL, Glaxo filters. We introduce penalties $P_i$ in the following way: the penalty for molecule equals 0 if the corresponding property $Pr_i$ lies in the acceptable range $[L_{min}, L_{max}]$ and linearly grows outside $P_i(M) = \text{ReLU}(L_{min} - Pr_i(M)) + \text{ReLU}(Pr_i(M) - L_{max})$. We consider the final reward to be a weighted sum of penalties and docking score: $r(s_t, a_t, s_{t+1}) = \max(0, DS(s_{t+1}) + \sum_i w_i P_i(s_{t+1}))$.

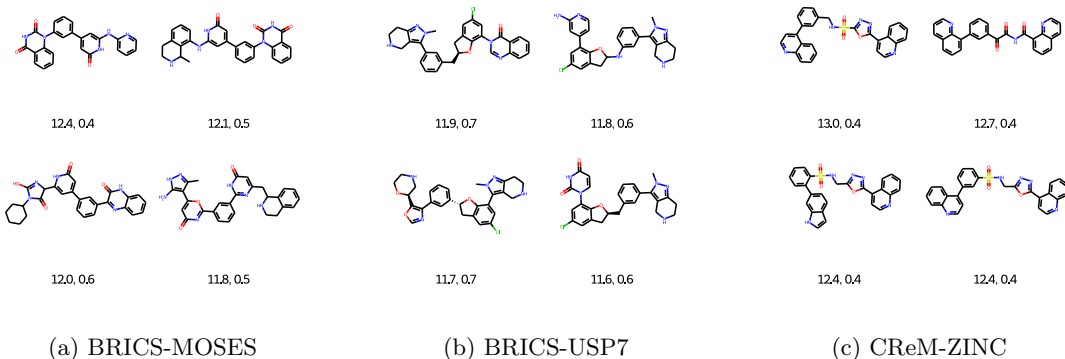

Figure 4: Selected representative molecules which were generated by FREED++ with USP7 as a target protein. Docking score and maximum Tanimoto similarity to set of known inhibitors depicted below molecules. Fragment libraries: BRICS-MOSES (A); BRICS-USP7 (B); CReM-ZINC (C).

**Results** The results of the FREED++ generation are highly dependent on the chosen fragment library. In the case of the BRICS USP7 fragment library, it is notable that FREED++-generated molecules are closely related to the previously reported USP7 inhibitors. The molecules generated using other libraries do not exhibit explicit structural analogy of the previously reported USP7 inhibitors while having higher docking scores. Such results indicate that FREED++ can design novel potential drug candidates and can serve as a motivation to test the inhibition activity of designed molecules experimentally.

## 6 Conclusions

In this paper, we present a detailed ablation study on the recent state-of-the-art objective-oriented molecular generation method FREED. Through extensive evaluation and deep analysis of FREED, we identify numerous implementation issues and inconsistencies in the evaluation protocol. We fix and simplify the original FREED model, provide an accurate comparison with an extended set of baselines, perform additional analysis on different fragment libraries, and test the approach in the practical scenario of USP7 inhibitors generation. We show that our FREED++ framework can produce molecules with superior docking scores compared to alternative approaches and can be a valuable tool for drug discovery.

**Code and Reproducibility**

The code for FFREED and FREED++ is accessible by link [7]. The code for other methods and all configs are available in supplementary material.

**Broader Impact Statement**

As our framework is a reimplementation of FREED, our "Broader Impact Statement" is similar to the original work. We provide a slightly edited version below.

FREED++ is a powerful tool that can generate various chemical compounds with desired properties. However, it is important to note that if this tool is used for malicious purposes, it can result in the creation of harmful substances such as biochemical weapons. The potential for abuse of this framework underscores the need for strict regulations and ethical guidelines to be put in place to prevent any misuse. Furthermore, it is crucial that users of this framework exercise caution and responsibility in their actions to ensure that it is only used for legitimate purposes that don't harm individuals or society as a whole.

---

[7]https://github.com/AIRI-Institute/FFREED

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

# A    Other changes

## A.1    Reproducibility

The docking score computation consists of 2 steps: first, a molecular conformation is generated from a molecular graph with OpenBabel software; then, the docking score is estimated with the QVina02 tool. Both steps include random sampling, thus requiring setting random seeds to ensure the reproducibility of the experiments. In the original implementation, seeds are set only for torch, random, numpy, and dgl packages while ignored for QVina02 and OpenBabel libraries. In our experiments, we fix that and set seeds for all the libraries.

## A.2    Fragment library preparation

For reward computation in FREED, molecular conformation is generated with OpenBabel software. In some cases, OpenBabel may fail to generate coordinates for charged molecules. We exclude charged fragments from the fragment library during the preprocessing step to avoid such situations. For the fragment library obtained with the CReM fragmentation method from the ZINC dataset, there are five charged fragments out of 91 total fragments. We use a filtered library of 86 uncharged fragments in the main experiments.

## A.3    Exploration

One commonly used strategy to increase agent exploration of the environment is to select actions uniformly. Such environment exploration can be done periodically with some probability ($\epsilon$-greedy strategies) or once before the actor and the critic updates. In the original implementation, the number of steps for uniform-random action selection was set to 3k, after which the real policy was run. While this technique helps exploration, we remove it from our framework for debugging purposes and to save computational time.

Additionally, we start an update of all networks' parameters as soon as the replay buffer is filled with batch-size observations. In the original implementation, the number of environment interactions to collect before starting to do gradient descent updates was set to 2k. Such a strategy is used to ensure that the replay buffer is complete enough for useful updates. However, we believe this stage should not change performance significantly, and we drop it for the simplicity of debugging.

We test the importance of described modifications on the DS optimization task. From table 8, we can see that additional exploration phase does not improve the performance of the model.

## A.4    Backbone update

To cope with the overestimation bias, the SAC algorithm uses an ensemble of two Q-functions (Haarnoja et al., 2018b). In the original FREED framework, the actor and both Q-functions share a GCN backbone, and the gradients are propagated to the backbone only through the critic loss. In the original implementation, the gradients are propagated only through one Q-function (a "stop gradient" is applied to the second one). We follow the original implementation of the SAC algorithm (Haarnoja et al., 2018b), and allow the gradients to propagate to the GCN through both Q-functions.

## A.5    Temperature parameter in SAC

FREED uses a version of the SAC algorithm with a learnable temperature parameter $\alpha$. Automatic tuning of $\alpha$ during training allows to avoid a costly manual grid-search. In the original implementation, $\alpha$ is tuned only between interaction steps 3000 and 33000. We do not see any reason for such a setup and follow the procedure described in (Haarnoja et al., 2018b), which suggests that alpha should be tuned throughout the whole training.

Moreover, the $\alpha$ parameter is clipped to be in the range $[0.05, 20]$ during the computation of the entropy loss term in the actor update. Here, we again follow (Haarnoja et al., 2018b) and discard the clipping of $\alpha$.

### A.6 Terminal states handling

In the FREED framework, trajectories are terminated either when the timelimit is reached or when there are no attachment points in the extended molecular graph representing the current state. Adding terminal states to the replay buffer is crucial, as the docking score can only be calculated in such states. In the original implementation, terminal states with no attachment points are not added to the replay buffer. This approach negatively influences the training of the critic as other states from such trajectories do not carry any information about the docking score.

### A.7 Reward shaping

In the original implementation, the authors use reward shaping (see eq. 15): at each step except for the terminal one, the agent received constant rewards $r_t = 0.05$. Typically, such constant reward is used when we want to increase episode length artificially. In FREED framework, the episode's length is fixed, so there is no point in such an increment.

Moreover, when the number of explicit atoms $NEA$ (including attachment points) in the molecular graph increases, an additional reward of 0.005 is given. Note hydrogen atoms considered to be presented in the graph implicitly, as in REINVENT, MolDQN, and GCPN. The only scenario in which $NEA(s_{t+1}) = NEA(s_t)$ is when an attachment point is replaced with a single atom-like Cl.

$$r(s_t, a_t, s_{t+1}) = \begin{cases} 0.05, & NEA(s_{t+1}) \leq NEA(s_t),\ t < T \\ 0.055, & NEA(s_{t+1}) > NEA(s_t),\ t < T \\ \max(0, DS(s_t)), & t = T. \end{cases} \tag{15}$$

In our experiments, we get rid of the reward shaping and only assign a non-zero reward to the terminal states, where it is possible to calculate the docking score:

$$r_t(s_t, a_t, s_{t+1}) = \begin{cases} 0, \text{ if } t < T; \\ \max(0, DS(s_{t+1})), \text{ if } t = T. \end{cases} \tag{16}$$

### A.8 Entropy estimation

As the SAC algorithm is designed to solve the Maximum Entropy RL task, the entropy regularization is crucial. To update both actor and critic in SAC, an estimate of the entropy $H[\pi_\phi]$ is used. A standart implementation of the algorithm (Haarnoja et al., 2018b) uses the following estimate:

$$H[\pi_\phi(s)] = -\log \pi_\phi(\widetilde{a_3}, \widetilde{a_2}, \widetilde{a_1}|s),\ \widetilde{a}_i \sim \pi_{\phi_i}\ i = 1, 2, 3$$

In FREED implementation, a non-default estimate of entropy is used:

$$\pi_\phi(a_3, a_2, a_1|\widetilde{a}_2, \widetilde{a}_1, s) = \pi_{\phi_3}(a_3|\widetilde{a}_2, \widetilde{a}_1, s)\ \pi_{\phi_2}(a_2|\widetilde{a}_1, s)\ \pi_{\phi_1}(a_1|s)$$

$$H[\pi_\phi(s)] = -\sum_{a_1, a_2, a_3} \pi_\phi(a_3, a_2, a_1|\widetilde{a}_2, \widetilde{a}_1, s) \log \pi_\phi(a_3, a_2, a_1|\widetilde{a}_2, \widetilde{a}_1, s),\ \widetilde{a}_1 \sim \pi_{\phi_1}, \widetilde{a}_2 \sim \pi_{\phi_2}$$

Despite the fact that such an estimate is unbiased, we use the vanilla version as it is easier to implement and the implementation is more readable.

### A.9 Arhitecture

Architectures of FREED, FFREED and FREED++ are presented in table 3. In fusing functions $\mathbf{f}_{\phi_1}, \mathbf{f}_{\phi_2}, \mathbf{f}_{\phi_3}$ for FFREED and FREED++ we use an output size $d_3$=128 instead $d_3$=64 in FREED. We use 3-layer MLPs

with a number of hidden neurons (128, 128) as projectors $\mathbf{f}_{\psi_1}, \mathbf{f}_{\psi_3}$, while in FREED, 2-layer MLPs with 32 neurons are used. Additionally, we use 4-layer MLPs as critic and auxiliary reward predictor heads with layer normalization. Moreover, in FREED, on the third step of action selection, embedding of fragment $\tilde{a}_2$ is used to condition the attachment point selection:

$$\tilde{A}_3(s, a_1, a_2) = \mathbf{f}_{\phi_3}(\tilde{F}_{a_2}, \tilde{F}_{a_2}^{\text{att}}) \tag{17}$$

We instead use fused representation of state, attachment point and fragment embeddings:

$$\tilde{A}_3(s, a_1, a_2) = \mathbf{f}_{\phi_3}(\tilde{a}_2, \tilde{F}_{a_2}^{\text{att}}) \tag{18}$$

To sum it up, we use a wider and deeper architecture for FFREED than the original FREED uses. In the case of FREED++, even with increased embedding sizes, the resulting architecture appears to be smaller (in terms of the total number of trainable parameters) because of the simplified mechanism of fragment selection (see section 4.3). To test the importance of the increased number of layers and neurons, we compare the FFREED model and its version FFREED(ORIG) with same module sizes as in FREED. We observe that this change does not affect the model performance (see table 4). Anyway, the resulting FREED++ model is faster and requires fewer parameters than FREED (see table 10).

| module | FREED | FFREED | FREED++ |
|---|---|---|---|
| $G_\theta$ | GCN(29, (128, 128, 128)) | GCN(29, (128, 128, 128)) | GCN(29, (128, 128, 128)) |
| $\mathbf{f}_{\phi_1}$ | MI(128, 128, 64) | MI(128, 128, 128) | CAT(128, 128, 128) |
| $\mathbf{f}_{\psi_1}$ | MLP(64, (32, 1)) | MLP(128, (128, 128, 1)) | MLP(128, (128, 128, 1)) |
| $\mathbf{f}_{\phi_2}$ | MI(1024, 64, 64) | MI(1024, 128, 128) | CAT(128, 128, 128) |
| $\mathbf{f}_{\psi_2}$ | MLP(64, (64, 64, 1)) | MLP(128, (128, 128, 1)) | MLP(128, (128, 128, 86)) |
| $\mathbf{f}_{\phi_3}$ | MI(128, 128, 64) | MI(128, 128, 128) | CAT(128, 128, 128) |
| $\mathbf{f}_{\psi_3}$ | MLP(64, (32, 1)) | MLP(128, (128, 128, 1)) | MLP(128, (128, 128, 1)) |
| $\mathbf{f}_\omega$ | MLP(299, (149, 1)) | MLP(512, (256, 128, 128, 1)) | MLP(512, (256, 128, 128, 1)) |
| $\mathbf{r}_\zeta^e$ | GCN(29, (128, 128, 128)) | GCN(29, (128, 128, 128)) | - |
| $\mathbf{r}_\zeta^h$ | MLP(128, (64, 1)) | MLP(128, (128, 128, 128, 1)) | - |

Table 3: Comparison of architecures of FREED, FFREED and FREED++.

## B  Gumbel-Softmax Issues

When sampling from categorical distribution $\pi$ defined by class probabilities $\pi_1, \ldots, \pi_k$, the following reparametrization is used (Jang et al., 2017):

$$y_i = \frac{\exp\left((\log \pi_i + g_i)/\tau\right)}{\sum_{j=1}^{k} \exp\left((\log \pi_j + g_j)/\tau\right)}, g_1, \ldots, g_k \text{ – i.i.d. from Gumbel(0, 1)}. \tag{19}$$

The authors of the original paper use the following reparametrization in their implementation:

$$
\begin{aligned}
y_i &= \frac{\exp\left((\pi_i + \nu g_i)/\tau\right)}{\sum_{j=1}^{k} \exp\left((\pi_j + \nu g_j)/\tau\right)} = \\
&= \frac{\exp\left((\frac{\pi_i}{\nu} + g_i)/\frac{\tau}{\nu}\right)}{\sum_{j=1}^{k} \exp\left((\frac{\pi_j}{\nu} + g_j)/\frac{\tau}{\nu}\right)} = \\
&= \frac{\exp\left((\log(\exp\frac{\pi_i}{\nu}) + g_i)/\frac{\tau}{\nu}\right)}{\sum_{j=1}^{k} \exp\left((\log(\exp\frac{\pi_j}{\nu}) + g_j)/\frac{\tau}{\nu}\right)}
\end{aligned}
\tag{20}
$$

As it can be seen from equation 20, sampling from such distribution is equivalent to sampling $\propto \exp(\frac{\pi}{\nu})$ with temperature $\frac{\pi}{\nu}$. In the original implementation, $\nu$ was set to $1 \times 10^{-3}$, and $\tau$ was set to $1 \times 10^{-1}$, making the temperature parameter equal to $\frac{\tau}{\nu} = 100$. The described issue had thus the following effect on the action distribution:

| protein | name | unique ($\uparrow$) | Avg DS ($\uparrow$) | Max DS ($\uparrow$) | Top-5 DS ($\uparrow$) |
|---|---|---|---|---|---|
| fa7 | FFREED(ORIG) | $0.60 \pm 0.18$ | $\mathbf{9.52 \pm 0.53}$ | $\mathbf{13.33 \pm 0.25}$ | $\mathbf{12.40 \pm 0.07}$ |
| | FFREED | $0.53 \pm 0.12$ | $8.39 \pm 0.82$ | $12.86 \pm 0.05$ | $12.37 \pm 0.01$ |
| | FREED | $0.43 \pm 0.40$ | $7.89 \pm 0.46$ | $10.00 \pm 1.12$ | $9.53 \pm 0.85$ |
| parp1 | FFREED(ORIG) | $0.58 \pm 0.17$ | $10.09 \pm 2.29$ | $17.50 \pm 0.91$ | $15.93 \pm 0.27$ |
| | FFREED | $0.58 \pm 0.09$ | $\mathbf{12.16 \pm 1.58}$ | $\mathbf{17.90 \pm 0.98}$ | $\mathbf{16.13 \pm 0.15}$ |
| | FREED | $0.33 \pm 0.32$ | $10.57 \pm 0.81$ | $12.80 \pm 1.91$ | $12.29 \pm 1.51$ |
| 5ht1b | FFREED(ORIG) | $0.50 \pm 0.20$ | $11.49 \pm 0.04$ | $14.60 \pm 0.79$ | $13.45 \pm 0.07$ |
| | FFREED* | $0.45 \pm 0.16$ | $\mathbf{11.97 \pm 0.20}$ | $\mathbf{14.85 \pm 0.21}$ | $\mathbf{13.99 \pm 0.45}$ |
| | FREED | $0.21 \pm 0.21$ | $6.43 \pm 5.57$ | $8.83 \pm 7.65$ | $7.97 \pm 6.90$ |
| fkb1a | FFREED(ORIG) | $0.42 \pm 0.09$ | $8.19 \pm 0.13$ | $\mathbf{11.50 \pm 0.00}$ | $\mathbf{10.65 \pm 0.26}$ |
| | FFREED | $0.36 \pm 0.14$ | $\mathbf{8.26 \pm 0.05}$ | $\mathbf{11.50 \pm 0.00}$ | $10.49 \pm 0.29$ |
| | FREED | $0.44 \pm 0.32$ | $5.71 \pm 0.92$ | $8.60 \pm 0.91$ | $8.00 \pm 0.69$ |
| abl1 | FFREED(ORIG) | $0.37 \pm 0.13$ | $10.74 \pm 0.35$ | $13.00 \pm 0.36$ | $\mathbf{12.39 \pm 0.19}$ |
| | FFREED | $0.39 \pm 0.12$ | $\mathbf{10.77 \pm 0.35}$ | $\mathbf{13.03 \pm 0.23}$ | $12.18 \pm 0.06$ |
| | FREED | $0.24 \pm 0.23$ | $2.17 \pm 1.68$ | $7.83 \pm 3.75$ | $6.36 \pm 5.28$ |
| usp7 | FFREED(ORIG) | $0.49 \pm 0.14$ | $10.79 \pm 0.50$ | $\mathbf{14.13 \pm 0.15}$ | $\mathbf{13.37 \pm 0.23}$ |
| | FFREED | $0.66 \pm 0.01$ | $\mathbf{10.91 \pm 0.15}$ | $14.03 \pm 0.46$ | $13.35 \pm 0.27$ |
| | FREED | $0.10 \pm 0.07$ | $8.58 \pm 1.02$ | $11.36 \pm 0.98$ | $11.12 \pm 0.88$ |

Table 4: Comparison of small and big FFREED-s. * denotes the model has a seed with Max DS $\leq 0$ on evaluation, while on the last epoch of training Min DS $> 0$. We exclude such runs.

1. The initial distribution was scaled by 1000, making some of the unnormalized probabilities $\gg 1$.

2. The exponent was applied to unnormalized probabilities, effectively making the distribution degenerate as the largest unnormalized probability dominated all the probability mass.

3. An extremely large temperature $\tau = 100$ was applied to somewhat counter the effect of the two previous steps.

Together with the apparent effect of making the action distribution deterministic and thus hindering the exploration, such reparametrization is inconsistent with the entropy optimization, as the entropy of the different distribution is maximized.

## C   Prioritized Experience Replay Issues

In this section, we discuss the PER proposed in the original paper. As mentioned in section 4.3, the probability of a transition being sampled from the replay buffer is defined by its novelty. Suppose that $\mathbf{r}_\zeta$ is the auxiliary reward model. Then, the unnormalized probabilities for transactions in the replay buffer are computed as follows:

$$p(s_t, a_t, s_{t+1}, r_t) = \begin{cases} |\mathbf{r}_\zeta(s_t) - \mathbf{f}_\omega(s_t, a_t)|, & t < T \\ |\mathbf{r}_\zeta(s_t) - r_t|, & t = T, \end{cases} \quad (21)$$

where $\mathbf{f}_\omega$ denotes the critic. The auxiliary reward model is trained to minimize the following objective:

$$\mathbb{E}_{(s_t, r_t) \sim \mathcal{D}_T} \left[ \|r_t - \mathbf{r}_\zeta(s_t)\|_2^2 \right] \to \min_\zeta, \quad (22)$$

where $\mathcal{D}_T$ denotes the set of all terminal states where reward $r_t \neq 0$.

Estimating the non-available reward in non-terminal states with a critic has several issues. The critic is trained to optimize the TD loss (see equation 12). The target for the critic includes a discounting factor $\gamma$

as well as an entropy term arising from the Maximum Entropy framework. Even if we consider $\gamma = 1$, as we operate with fixed-length episodes, the entropy term in the critic target makes it incorrect to use the critic's output as a reward estimate. We hypothesize that for this exact reason, the entropy term was omitted from the critic target in the original implementation (see section 4.1).

## D   Comparison with baselines

| protein | $\sigma$ | unique ($\uparrow$) | Avg DS ($\uparrow$) | Max DS ($\uparrow$) | Top-5 DS ($\uparrow$) |
|---------|----------|---------------------|---------------------|---------------------|------------------------|
| fa7 | 60 | $0.99 \pm 0.00$ | $5.98 \pm 4.73$ | $10.96 \pm 0.72$ | $9.80 \pm 0.88$ |
| | 100 | $0.99 \pm 0.00$ | $\mathbf{8.79 \pm 0.70}$ | $\mathbf{11.23 \pm 0.45}$ | $\mathbf{10.49 \pm 0.48}$ |
| | 200 | $0.90 \pm 0.04$ | $2.86 \pm 3.41$ | $10.23 \pm 0.80$ | $6.71 \pm 5.46$ |
| usp7 | 60 | $0.99 \pm 0.00$ | $9.69 \pm 0.39$ | $\mathbf{13.13 \pm 0.89}$ | $\mathbf{12.05 \pm 0.73}$ |
| | 100 | $0.97 \pm 0.01$ | $9.70 \pm 0.23$ | $11.56 \pm 0.20$ | $10.99 \pm 0.05$ |
| | 200 | $0.79 \pm 0.04$ | $\mathbf{10.26 \pm 1.31}$ | $11.93 \pm 1.17$ | $11.50 \pm 1.35$ |
| parp1 | 60 | $0.98 \pm 0.02$ | $\mathbf{10.04 \pm 1.42}$ | $\mathbf{15.93 \pm 0.15}$ | $\mathbf{14.06 \pm 0.81}$ |
| | 100 | $0.96 \pm 0.02$ | $9.28 \pm 1.62$ | $14.33 \pm 0.46$ | $13.35 \pm 0.75$ |
| | 200 | $0.64 \pm 0.12$ | $8.90 \pm 3.59$ | $14.19 \pm 1.90$ | $13.20 \pm 1.18$ |
| 5ht1b | 60 | $0.99 \pm 0.01$ | $10.27 \pm 0.16$ | $\mathbf{13.33 \pm 0.11}$ | $\mathbf{12.20 \pm 0.27}$ |
| | 100 | $0.99 \pm 0.00$ | $8.71 \pm 3.28$ | $13.06 \pm 0.90$ | $12.08 \pm 0.91$ |
| | 200 | $0.90 \pm 0.13$ | $\mathbf{10.43 \pm 2.12}$ | $12.13 \pm 2.55$ | $11.58 \pm 2.47$ |
| abl1 | 60 | $0.98 \pm 0.00$ | $10.08 \pm 0.95$ | $12.40 \pm 0.95$ | $11.66 \pm 1.21$ |
| | 100 | $0.97 \pm 0.00$ | $10.86 \pm 0.66$ | $12.83 \pm 0.60$ | $12.21 \pm 0.47$ |
| | 200 | $0.85 \pm 0.03$ | $\mathbf{11.61 \pm 0.57}$ | $\mathbf{13.16 \pm 0.20}$ | $\mathbf{12.56 \pm 0.42}$ |
| fkb1a | 60 | $0.99 \pm 0.00$ | $7.16 \pm 0.23$ | $9.03 \pm 0.37$ | $8.37 \pm 0.36$ |
| | 100 | $0.98 \pm 0.01$ | $7.20 \pm 0.12$ | $8.90 \pm 0.36$ | $8.35 \pm 0.13$ |
| | 200 | $0.93 \pm 0.03$ | $\mathbf{7.83 \pm 0.21}$ | $\mathbf{9.33 \pm 0.11}$ | $\mathbf{8.88 \pm 0.05}$ |

Table 5: Comparison of different $\sigma$ parameters for REINVENT on DS optimization task.

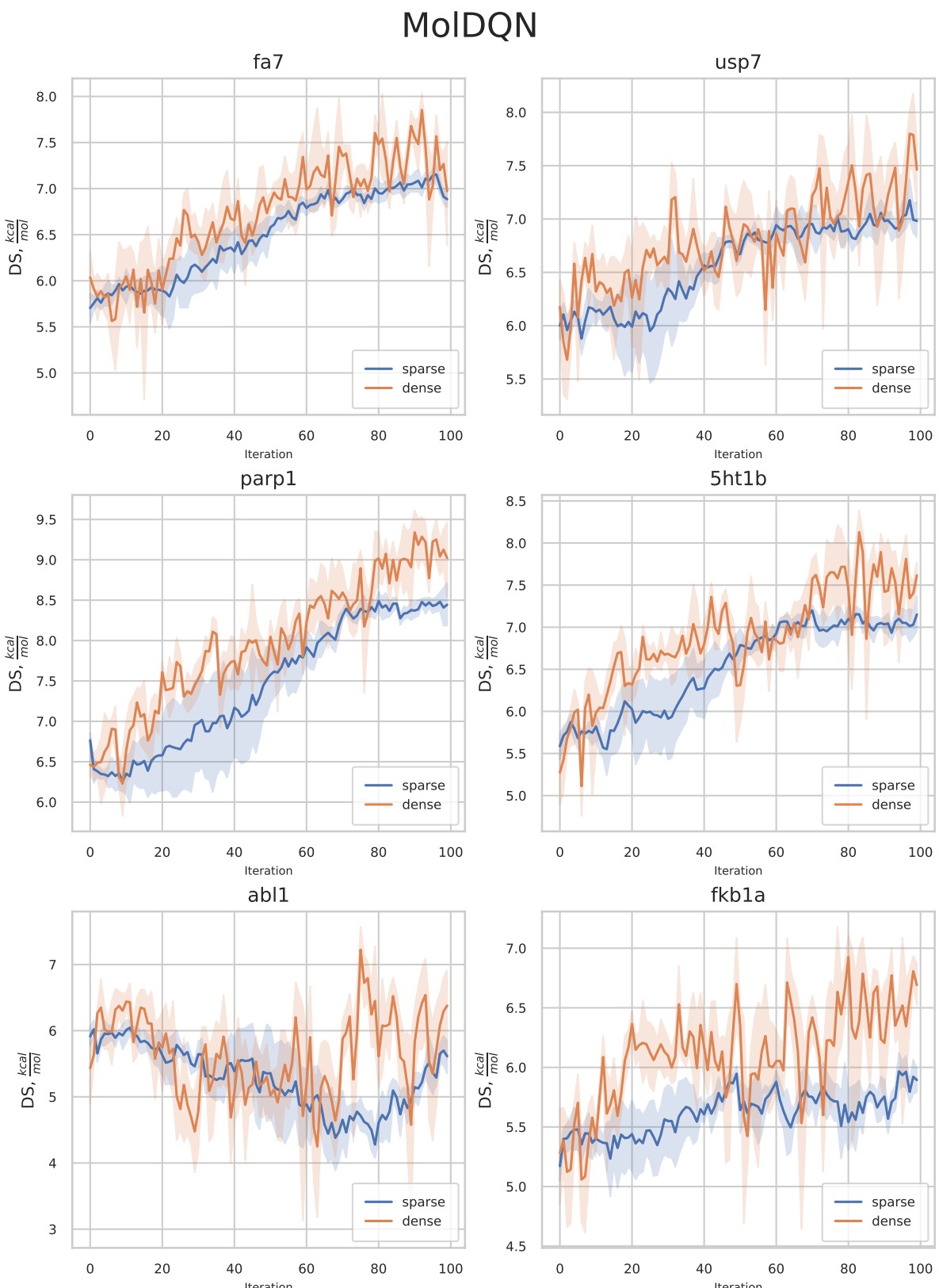

Figure 5: Average docking score over batch during training for MolDQN. Shaded regions denote 95% confidence interval.

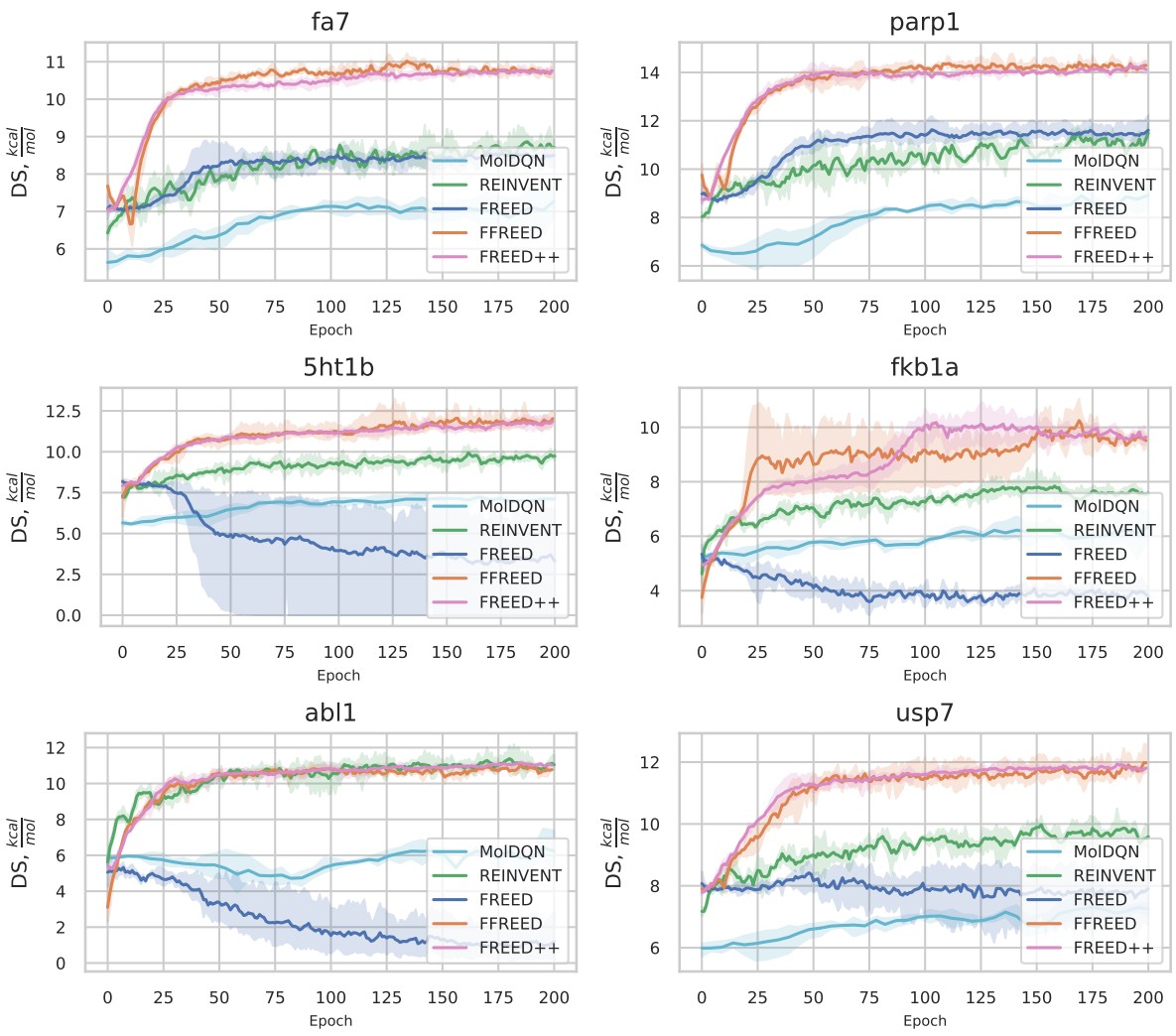

Figure 6: Average docking score over batch during training. Training curves aligned on x-axis. Shaded regions denote 95% confidence interval.

| protein | method | unique (↑) | Avg DS (↑) | Max DS (↑) | Top-5 DS (↑) |
|---------|--------|-----------|-----------|-----------|-------------|
| fa7 | CombGen | $0.98 \pm 0.00$ | $7.15 \pm 0.01$ | $12.00 \pm 0.30$ | $9.40 \pm 0.01$ |
| | MolDQN | $0.19 \pm 0.01$ | $8.00 \pm 0.27$ | $9.93 \pm 0.15$ | $9.57 \pm 0.18$ |
| | REINVENT | $0.99 \pm 0.00$ | $8.79 \pm 0.70$ | $11.23 \pm 0.45$ | $10.49 \pm 0.48$ |
| | Pocket2Mol | $1.00 \pm 0.00$ | $8.26 \pm 0.01$ | $11.16 \pm 0.20$ | $10.24 \pm 0.26$ |
| | FREED | $0.43 \pm 0.40$ | $7.89 \pm 0.46$ | $10.00 \pm 1.12$ | $9.53 \pm 0.85$ |
| | FFREED | $0.53 \pm 0.12$ | $8.39 \pm 0.82$ | $12.86 \pm 0.05$ | $\mathbf{12.37 \pm 0.01}$ |
| | FREED++ | $0.76 \pm 0.05$ | $7.78 \pm 1.80$ | $\mathbf{13.23 \pm 0.41}$ | $12.37 \pm 0.07$ |
| | KI | - | $\mathbf{8.85}$ | $10.30$ | $10.10$ |
| parp1 | CombGen | $0.98 \pm 0.00$ | $8.50 \pm 0.01$ | $15.26 \pm 0.57$ | $11.82 \pm 0.00$ |
| | MolDQN | $0.22 \pm 0.02$ | $9.51 \pm 0.49$ | $11.89 \pm 0.20$ | $11.35 \pm 0.30$ |
| | REINVENT | $0.98 \pm 0.02$ | $10.04 \pm 1.42$ | $15.93 \pm 0.15$ | $14.06 \pm 0.81$ |
| | Pocket2Mol | $1.00 \pm 0.00$ | $11.16 \pm 0.13$ | $15.66 \pm 0.98$ | $14.09 \pm 0.54$ |
| | FREED | $0.33 \pm 0.32$ | $10.57 \pm 0.81$ | $12.80 \pm 1.91$ | $12.29 \pm 1.51$ |
| | FFREED | $0.58 \pm 0.09$ | $12.16 \pm 1.58$ | $\mathbf{17.90 \pm 0.98}$ | $16.13 \pm 0.15$ |
| | FREED++ | $0.71 \pm 0.06$ | $\mathbf{12.98 \pm 0.15}$ | $17.73 \pm 0.83$ | $\mathbf{16.26 \pm 0.19}$ |
| | KI | - | $10.13$ | $13.60$ | $12.51$ |
| 5ht1b | CombGen | $0.98 \pm 0.00$ | $7.63 \pm 0.04$ | $13.13 \pm 0.20$ | $10.85 \pm 0.01$ |
| | MolDQN | $0.20 \pm 0.03$ | $7.97 \pm 0.10$ | $10.36 \pm 0.64$ | $9.61 \pm 0.17$ |
| | REINVENT | $0.99 \pm 0.01$ | $10.27 \pm 0.16$ | $13.33 \pm 0.11$ | $12.20 \pm 0.27$ |
| | Pocket2Mol | $1.00 \pm 0.00$ | $6.23 \pm 0.56$ | $13.26 \pm 0.87$ | $11.72 \pm 0.33$ |
| | FREED | $0.21 \pm 0.21$ | $6.43 \pm 5.57$ | $8.83 \pm 7.65$ | $7.97 \pm 6.90$ |
| | FFREED* | $0.45 \pm 0.16$ | $\mathbf{11.97 \pm 0.20}$ | $\mathbf{14.85 \pm 0.21}$ | $\mathbf{13.99 \pm 0.45}$ |
| | FREED++ | $0.45 \pm 0.07$ | $11.78 \pm 0.12$ | $14.50 \pm 0.79$ | $13.50 \pm 0.15$ |
| | KI | - | $8.07$ | $10.60$ | $10.52$ |
| fkb1a | CombGen | $0.98 \pm 0.00$ | $3.99 \pm 0.10$ | $9.56 \pm 0.15$ | $7.94 \pm 0.01$ |
| | MolDQN | $0.18 \pm 0.03$ | $6.75 \pm 0.16$ | $8.79 \pm 0.17$ | $8.20 \pm 0.12$ |
| | REINVENT | $0.93 \pm 0.03$ | $7.83 \pm 0.21$ | $9.33 \pm 0.11$ | $8.88 \pm 0.05$ |
| | Pocket2Mol | $1.00 \pm 0.00$ | $5.75 \pm 0.03$ | $9.00 \pm 0.45$ | $8.22 \pm 0.30$ |
| | FREED | $0.44 \pm 0.32$ | $5.71 \pm 0.92$ | $8.60 \pm 0.91$ | $8.00 \pm 0.69$ |
| | FFREED | $0.36 \pm 0.14$ | $8.26 \pm 0.05$ | $\mathbf{11.50 \pm 0.00}$ | $10.49 \pm 0.29$ |
| | FREED++ | $0.30 \pm 0.03$ | $\mathbf{8.50 \pm 0.06}$ | $\mathbf{11.50 \pm 0.00}$ | $\mathbf{10.92 \pm 0.02}$ |
| | KI | - | $6.99$ | $8.90$ | $8.76$ |
| abl1 | CombGen | $0.98 \pm 0.00$ | $3.77 \pm 0.17$ | $12.83 \pm 0.25$ | $10.57 \pm 0.00$ |
| | MolDQN | $0.16 \pm 0.02$ | $6.99 \pm 0.43$ | $9.50 \pm 0.36$ | $9.05 \pm 0.49$ |
| | REINVENT | $0.85 \pm 0.03$ | $\mathbf{11.61 \pm 0.57}$ | $13.16 \pm 0.20$ | $\mathbf{12.56 \pm 0.42}$ |
| | Pocket2Mol | $1.00 \pm 0.00$ | $3.27 \pm 0.14$ | $\mathbf{13.26 \pm 0.72}$ | $11.37 \pm 0.28$ |
| | FREED | $0.24 \pm 0.23$ | $2.17 \pm 1.68$ | $7.83 \pm 3.75$ | $6.36 \pm 5.28$ |
| | FFREED | $0.39 \pm 0.12$ | $10.77 \pm 0.35$ | $13.03 \pm 0.23$ | $12.18 \pm 0.06$ |
| | FREED++ | $0.34 \pm 0.02$ | $11.00 \pm 0.10$ | $12.86 \pm 0.05$ | $12.36 \pm 0.21$ |
| | KI | - | $3.02$ | $11.70$ | $10.36$ |
| usp7 | CombGen | $0.98 \pm 0.00$ | $8.03 \pm 0.00$ | $13.00 \pm 0.26$ | $10.53 \pm 0.00$ |
| | MolDQN | $0.21 \pm 0.03$ | $7.92 \pm 0.23$ | $9.93 \pm 0.41$ | $9.43 \pm 0.34$ |
| | REINVENT | $0.99 \pm 0.00$ | $9.69 \pm 0.39$ | $13.13 \pm 0.89$ | $12.05 \pm 0.73$ |
| | Pocket2Mol | $1.00 \pm 0.00$ | $8.71 \pm 0.37$ | $12.36 \pm 0.40$ | $11.30 \pm 0.65$ |
| | FREED | $0.10 \pm 0.07$ | $8.58 \pm 1.02$ | $11.36 \pm 0.98$ | $11.12 \pm 0.88$ |
| | FFREED | $0.66 \pm 0.01$ | $\mathbf{10.91 \pm 0.15}$ | $14.03 \pm 0.46$ | $\mathbf{13.35 \pm 0.27}$ |
| | FREED++ | $0.64 \pm 0.07$ | $9.66 \pm 3.28$ | $\mathbf{14.16 \pm 0.47}$ | $13.30 \pm 0.13$ |
| | KI | - | $9.40$ | $11.90$ | $11.90$ |

Table 6: Comparison of FREED, FFREED and FREED++ with baselines on DS optimization task. * denotes the model has a seed with Max DS $\leq 0$ on evaluation, while on the last epoch of training Min DS $> 0$. We exclude such runs.

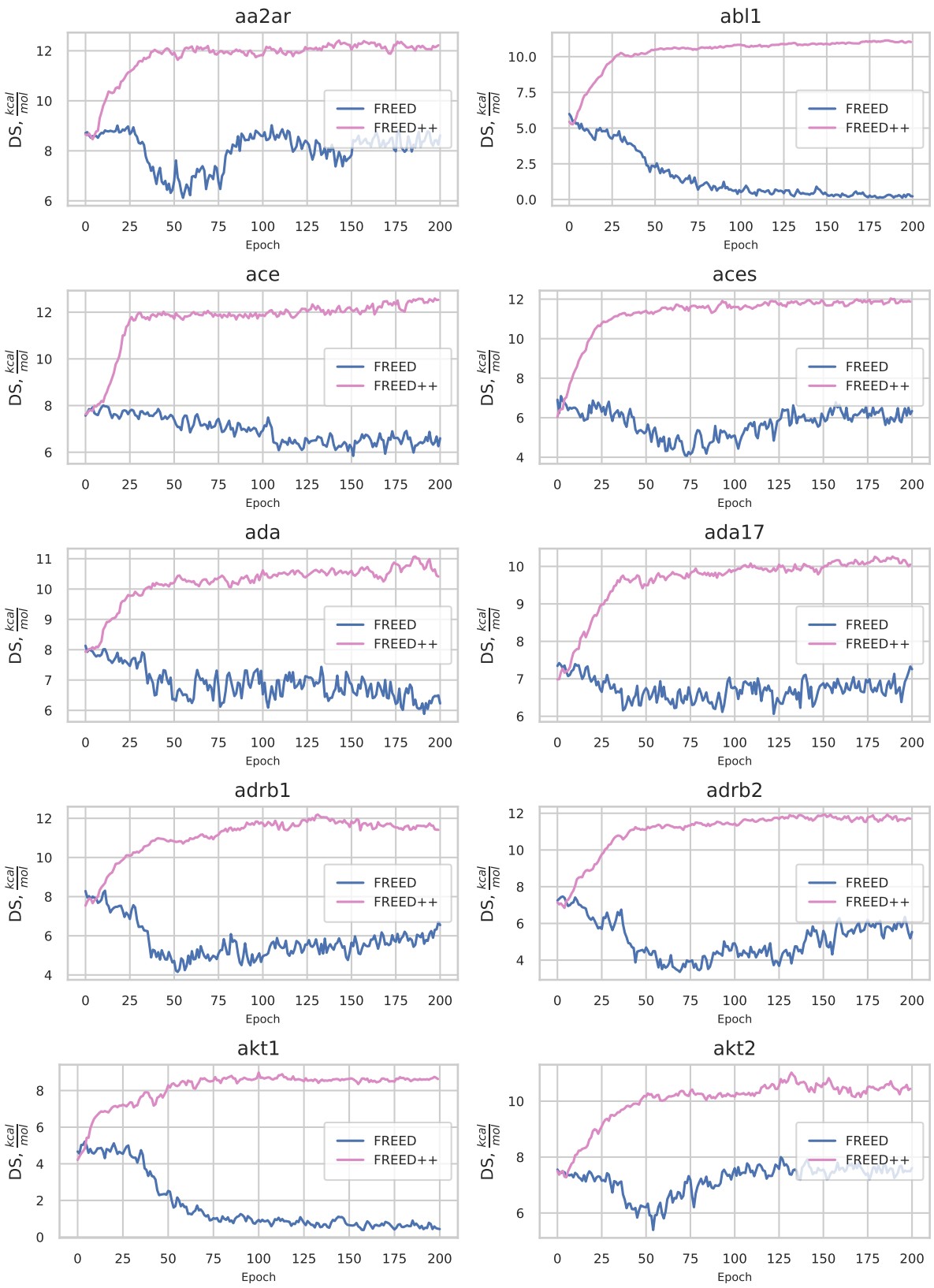

Figure 7: Comparison of FREED and FREED++ on DS optimization task. 10 first proteins from DUDE considered as targets.

# E    Fragments library experiments

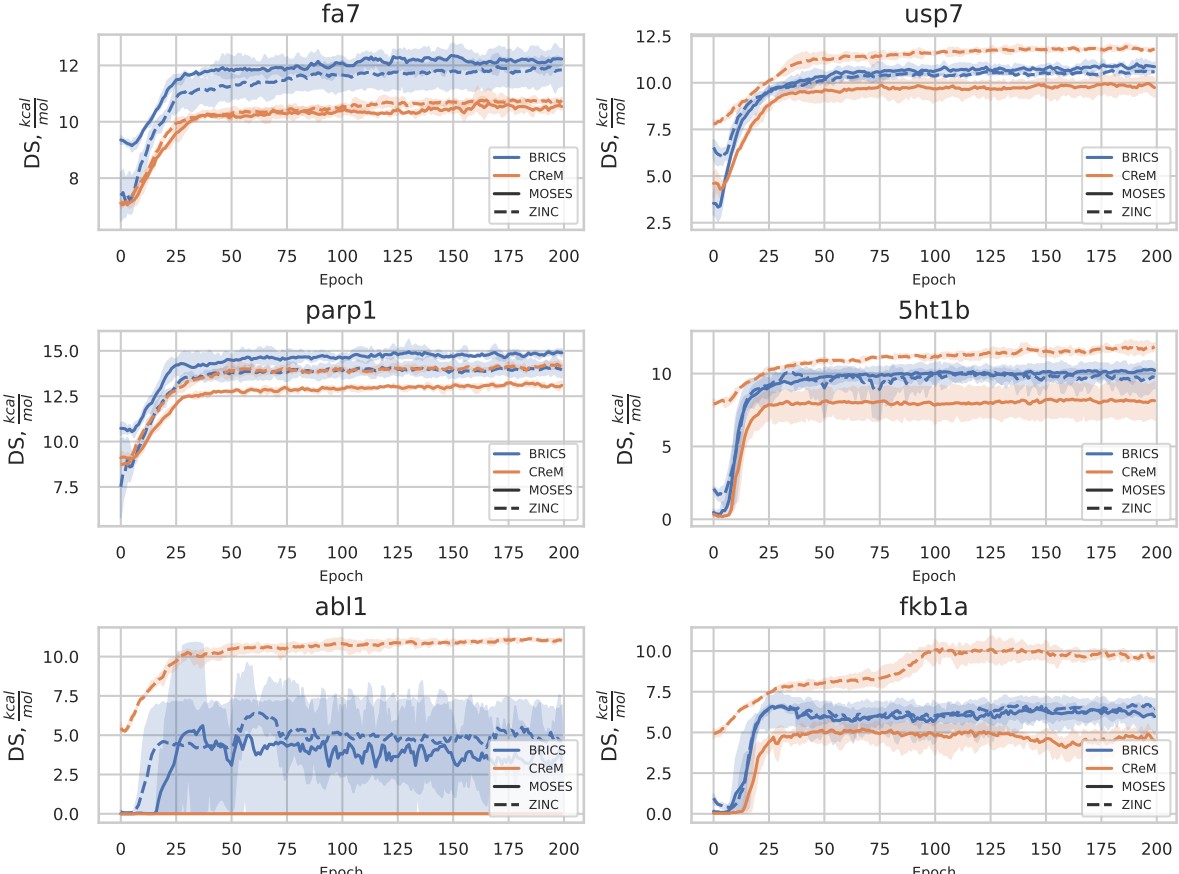

Figure 8: Average docking score over batch during training. Shaded regions denote 95% confidence interval.

# F    Simplifications of FFREED

In this section, we compare FFREED, FREED++ models and their intermediate versions. As described in 4, FREED++ differs from FFREED in 1) the absence of prioritization, 2) the simplified mechanism of fragment selection, and 3) the usage of "lightweight" fusing functions. All proposed modifications are aimed to decrease the total amount of parameters in the model and speed up the training process.

We perform ablation studies on the DS optimization task to test the importance of described modifications. We dub FREED++ with a prioritization mechanism FREED++(PER), FREED++ with a mechanism of fragment selection taken from FFREED as FREED++(AM), and FREED++ with multiplicative interactions as fusing functions as FREED++(FUSE). As it can be seen in table 9, the overall performance of different models is approximately the same, while it can differ significantly for particular targets. For example, FREED++ stably outperforms FREED++(PER) for fa7 protein, while the situation is inversed for parp1 target. The same effect can be seen for FREED++ and FREED++(FUSE) models on fa7 and parp1 proteins.

To compare the speed of different FFREED versions, we collect $10^4$ transactions with a uniform policy, split them into 100 batches, and measure the time of gradient updates for each method. From table 10, we can see that for all modifications of FREED++ in the direction of FFREED number of trainable parameters is increased. The most significant increase ($\sim$7.3x) occurs when MI is used as a fusing function. Gradient update times increase when adding modifications to FREED++ only for PER and FUSE settings; for AM

| protein | fragmentation | dataset | unique (↑) | Avg DS (↑) | Max DS (↑) | Top-5 DS (↑) |
|---|---|---|---|---|---|---|
| 5ht1b | BRICS | MOSES | 0.56 ± 0.16 | 9.28 ± 2.22 | 13.40 ± 0.65 | 12.44 ± 0.32 |
| | | ZINC | 0.47 ± 0.10 | 9.47 ± 1.65 | 13.03 ± 0.20 | 12.23 ± 0.31 |
| | CReM | MOSES | 0.69 ± 0.18 | 7.07 ± 1.90 | 11.86 ± 0.61 | 10.79 ± 0.60 |
| | | ZINC | 0.45 ± 0.07 | **11.78 ± 0.12** | **14.50 ± 0.79** | **13.50 ± 0.15** |
| abl1 | BRICS | MOSES | 0.52 ± 0.41 | 1.29 ± 1.31 | 7.06 ± 6.23 | 6.33 ± 5.58 |
| | | ZINC | 0.61 ± 0.13 | 5.02 ± 3.45 | 9.56 ± 2.18 | 8.78 ± 2.50 |
| | CReM | MOSES | 1.00 ± 0.00 | 0.00 ± 0.00 | 0.00 ± 0.00 | 0.00 ± 0.00 |
| | | ZINC | 0.34 ± 0.02 | **11.00 ± 0.10** | **12.86 ± 0.05** | **12.36 ± 0.21** |
| fa7 | BRICS | MOSES | 0.61 ± 0.09 | 3.10 ± 0.26 | **14.20 ± 0.34** | **13.09 ± 0.07** |
| | | ZINC | 0.66 ± 0.14 | 2.82 ± 0.81 | 13.66 ± 0.94 | 12.48 ± 0.89 |
| | CReM | MOSES | 0.64 ± 0.13 | 1.29 ± 0.95 | 12.56 ± 0.40 | 9.26 ± 3.78 |
| | | ZINC | 0.76 ± 0.05 | **7.78 ± 1.80** | 13.23 ± 0.41 | 12.37 ± 0.07 |
| fkb1a | BRICS | MOSES | 0.57 ± 0.13 | 5.79 ± 1.65 | 9.03 ± 0.35 | 8.35 ± 0.37 |
| | | ZINC | 0.47 ± 0.36 | 6.55 ± 0.86 | 9.86 ± 0.46 | 9.20 ± 0.33 |
| | CReM | MOSES | 0.57 ± 0.15 | 4.93 ± 0.68 | 8.26 ± 0.68 | 7.67 ± 0.64 |
| | | ZINC | 0.30 ± 0.03 | **8.50 ± 0.06** | **11.50 ± 0.00** | **10.92 ± 0.02** |
| parp1 | BRICS | MOSES | 0.51 ± 0.13 | 5.89 ± 1.83 | 16.96 ± 0.49 | 16.07 ± 0.31 |
| | | ZINC | 0.60 ± 0.27 | 6.82 ± 1.60 | 16.70 ± 0.36 | 15.51 ± 0.53 |
| | CReM | MOSES | 0.70 ± 0.03 | 1.51 ± 0.86 | 15.69 ± 0.50 | 13.82 ± 0.57 |
| | | ZINC | 0.71 ± 0.06 | **12.98 ± 0.15** | **17.73 ± 0.83** | **16.26 ± 0.19** |
| usp7 | BRICS | MOSES | 0.58 ± 0.17 | 5.82 ± 4.32 | 13.16 ± 0.51 | 12.22 ± 0.67 |
| | | ZINC | 0.63 ± 0.13 | 9.07 ± 0.70 | 13.39 ± 0.88 | 12.42 ± 0.46 |
| | CReM | MOSES | 0.74 ± 0.07 | 1.21 ± 1.06 | 12.00 ± 1.81 | 7.77 ± 6.27 |
| | | ZINC | 0.64 ± 0.07 | **9.66 ± 3.28** | **14.16 ± 0.47** | **13.30 ± 0.13** |

Table 7: Comparison of different fragments libraries used in FREED++ on DS optimization task.

| protein | method | unique (↑) | Avg DS (↑) | Max DS (↑) | Top-5 DS (↑) |
|---|---|---|---|---|---|
| fa7 | FREED++ | 0.76 ± 0.05 | 7.78 ± 1.80 | **13.23 ± 0.41** | **12.37 ± 0.07** |
| | FREED++(EXPL) | 0.62 ± 0.05 | **8.84 ± 1.53** | 13.03 ± 0.11 | 12.36 ± 0.11 |
| parp1 | FREED++ | 0.71 ± 0.06 | **12.98 ± 0.15** | **17.73 ± 0.83** | **16.26 ± 0.19** |
| | FREED++(EXPL) | 0.67 ± 0.17 | 11.37 ± 1.72 | 17.26 ± 0.73 | 16.04 ± 0.11 |
| 5ht1b | FREED++ | 0.45 ± 0.07 | 11.78 ± 0.12 | 14.50 ± 0.79 | 13.50 ± 0.15 |
| | FREED++(EXPL) | 0.48 ± 0.04 | **11.98 ± 0.14** | **14.80 ± 0.34** | **13.68 ± 0.15** |
| fkb1a | FREED++ | 0.30 ± 0.03 | **8.50 ± 0.06** | **11.50 ± 0.00** | **10.92 ± 0.02** |
| | FREED++(EXPL) | 0.39 ± 0.18 | 8.24 ± 0.20 | **11.50 ± 0.00** | 10.29 ± 0.55 |
| abl1 | FREED++ | 0.34 ± 0.02 | 11.00 ± 0.10 | 12.86 ± 0.05 | **12.36 ± 0.21** |
| | FREED++(EXPL) | 0.31 ± 0.03 | **11.09 ± 0.03** | **12.93 ± 0.32** | 12.19 ± 0.15 |
| usp7 | FREED++ | 0.64 ± 0.07 | 9.66 ± 3.28 | 14.16 ± 0.47 | **13.30 ± 0.13** |
| | FREED++(EXPL) | 0.61 ± 0.16 | **11.15 ± 0.42** | **14.26 ± 0.30** | 13.19 ± 0.07 |

Table 8: Comparison of FREED++ run with additional exploration stage and without.

version, update time stays approximately the same. FREED has the worst performance in terms of size and time required for a gradient update - FFREED is around ∼8.5x slower on average and ∼22x bigger than FREED++.

| protein | version | unique (↑) | Avg DS (↑) | Max DS (↑) | Top-5 DS (↑) |
|---|---|---|---|---|---|
| fa7 | FREED++ | 0.76 ± 0.05 | 7.78 ± 1.80 | 13.23 ± 0.41 | 12.37 ± 0.07 |
| | FFREED | 0.53 ± 0.12 | **8.39 ± 0.82** | 12.86 ± 0.05 | 12.37 ± 0.01 |
| | FREED++(PER)* | 0.65 ± 0.05 | 6.24 ± 0.85 | 12.90 ± 0.00 | 12.19 ± 0.18 |
| | FREED++(AM) | 0.69 ± 0.19 | 8.14 ± 0.90 | 12.96 ± 0.20 | 12.28 ± 0.04 |
| | FREED++(FUSE) | 0.48 ± 0.02 | 8.31 ± 0.98 | **13.26 ± 0.30** | **12.66 ± 0.18** |
| parp1 | FREED++ | 0.71 ± 0.06 | 12.98 ± 0.15 | 17.73 ± 0.83 | 16.26 ± 0.19 |
| | FFREED | 0.58 ± 0.09 | 12.16 ± 1.58 | 17.90 ± 0.98 | 16.13 ± 0.15 |
| | FREED++(PER) | 0.64 ± 0.07 | **13.25 ± 0.53** | 18.16 ± 0.75 | **16.46 ± 0.03** |
| | FREED++(AM) | 0.59 ± 0.21 | 12.36 ± 1.09 | **18.20 ± 0.86** | 16.27 ± 0.12 |
| | FREED++(FUSE) | 0.73 ± 0.06 | 9.63 ± 3.01 | 16.89 ± 0.43 | 15.81 ± 0.31 |
| 5ht1b | FREED++ | 0.45 ± 0.07 | 11.78 ± 0.12 | 14.50 ± 0.79 | 13.50 ± 0.15 |
| | FFREED* | 0.45 ± 0.16 | **11.97 ± 0.20** | **14.85 ± 0.21** | **13.99 ± 0.45** |
| | FREED++(PER) | 0.56 ± 0.10 | 11.78 ± 0.21 | 14.73 ± 0.55 | 13.65 ± 0.30 |
| | FREED++(AM)* | 0.45 ± 0.08 | 11.94 ± 0.13 | 14.00 ± 0.00 | 13.42 ± 0.16 |
| | FREED++(FUSE) | 0.55 ± 0.05 | 11.54 ± 0.14 | 14.46 ± 0.72 | 13.38 ± 0.08 |

Table 9: Comparison of different version of FFREED on DS optimization task. * denotes the model has a seed with Max DS ≤ 0 on evaluation, while on the last epoch of training Min DS > 0. We exclude such runs.

| method | size, MB | time, sec |
|---|---|---|
| FREED | 24.12 | 1.21 ± 0.30 |
| FREED++ | 3.77 | 0.38 ± 0.13 |
| FFREED | 84.60 | 3.28 ± 0.47 |
| FREED++(PER) | 4.17 | 0.52 ± 0.21 |
| FREED++(AM) | 4.18 | 0.36 ± 0.12 |
| FREED++(FUSE) | 27.80 | 0.48 ± 0.11 |

Table 10: Comparison of the speed of different versions of FFREED. Time denotes gradient update time measured over a batch with size 100.

## G   Fragments library preprocessing

One valuable property of FREED on which authors of the original paper were focused is the ability to incorporate prior knowledge into generation via filtration of used fragments. For example, fragments which not passed commonly used medicinal chemistry filters can be excluded during the preprocessing of fragments collection.

We compare two generation setups: with and without filtration of the fragments dictionary. In experiments, we use the same set of fragments as in section 5.1 CReM/ZINC. Fragments were filtered with Glaxo  (Lane et al., 2006), SureChEMBL  (Papadatos et al., 2015), and PAINS  (Baell & Holloway, 2010) filters.

| protein | method | library | unique (↑) | Avg DS (↑) | Max DS (↑) | Top-5 DS (↑) | PAINS (↑) | SureChEMBL (↑) | Glaxo (↑) |
|---|---|---|---|---|---|---|---|---|---|
| fa7 | FREED++ | filtered | $0.62 \pm 0.10$ | $\mathbf{10.49 \pm 0.13}$ | $13.16 \pm 0.35$ | $\mathbf{12.45 \pm 0.15}$ | $\mathbf{0.99 \pm 0.00}$ | $0.96 \pm 0.01$ | $0.99 \pm 0.00$ |
| | | vanilla | $0.76 \pm 0.05$ | $7.78 \pm 1.80$ | $\mathbf{13.23 \pm 0.41}$ | $12.37 \pm 0.07$ | $0.33 \pm 0.21$ | $\mathbf{0.97 \pm 0.00}$ | $0.99 \pm 0.00$ |
| | FREED | filtered | $0.24 \pm 0.41$ | $8.91 \pm 0.26$ | $11.50 \pm 0.34$ | $9.45 \pm 1.19$ | $\mathbf{0.99 \pm 0.00}$ | $0.95 \pm 0.08$ | $\mathbf{1.00 \pm 0.00}$ |
| | | vanilla | $0.43 \pm 0.40$ | $7.89 \pm 0.46$ | $10.00 \pm 1.12$ | $9.53 \pm 0.85$ | $0.97 \pm 0.02$ | $0.77 \pm 0.24$ | $0.74 \pm 0.23$ |
| parp1 | FREED++ | filtered | $0.56 \pm 0.09$ | $12.23 \pm 1.19$ | $\mathbf{18.00 \pm 1.10}$ | $\mathbf{16.50 \pm 0.50}$ | $0.99 \pm 0.00$ | $\mathbf{0.97 \pm 0.00}$ | $0.99 \pm 0.00$ |
| | | vanilla | $0.71 \pm 0.06$ | $\mathbf{12.98 \pm 0.15}$ | $17.73 \pm 0.83$ | $16.26 \pm 0.19$ | $0.45 \pm 0.27$ | $\mathbf{0.97 \pm 0.00}$ | $0.98 \pm 0.01$ |
| | FREED | filtered | $0.04 \pm 0.06$ | $12.35 \pm 0.91$ | $15.20 \pm 0.70$ | $13.18 \pm 2.27$ | $\mathbf{1.00 \pm 0.00}$ | $0.97 \pm 0.03$ | $\mathbf{1.00 \pm 0.00}$ |
| | | vanilla | $0.33 \pm 0.32$ | $10.57 \pm 0.81$ | $12.80 \pm 1.91$ | $12.29 \pm 1.51$ | $0.95 \pm 0.02$ | $0.89 \pm 0.12$ | $0.91 \pm 0.09$ |
| 5ht1b | FREED++ | filtered | $0.31 \pm 0.01$ | $11.46 \pm 0.10$ | $13.36 \pm 0.28$ | $12.95 \pm 0.26$ | $0.99 \pm 0.00$ | $0.88 \pm 0.05$ | $\mathbf{0.99 \pm 0.00}$ |
| | | vanilla | $0.45 \pm 0.07$ | $\mathbf{11.78 \pm 0.12}$ | $\mathbf{14.50 \pm 0.79}$ | $\mathbf{13.50 \pm 0.15}$ | $0.97 \pm 0.00$ | $\mathbf{0.89 \pm 0.04}$ | $\mathbf{0.99 \pm 0.00}$ |
| | FREED | filtered | $0.28 \pm 0.27$ | $7.50 \pm 4.55$ | $10.06 \pm 4.82$ | $8.90 \pm 5.74$ | $\mathbf{0.99 \pm 0.00}$ | $0.83 \pm 0.18$ | $\mathbf{0.99 \pm 0.00}$ |
| | | vanilla | $0.21 \pm 0.21$ | $6.43 \pm 5.57$ | $8.83 \pm 7.65$ | $7.97 \pm 6.90$ | $0.63 \pm 0.54$ | $0.56 \pm 0.50$ | $0.60 \pm 0.53$ |
| fkb1a | FREED++ | filtered | $0.32 \pm 0.04$ | $8.39 \pm 0.36$ | $10.30 \pm 1.05$ | $9.81 \pm 1.00$ | $\mathbf{0.99 \pm 0.00}$ | $\mathbf{0.86 \pm 0.05}$ | $0.98 \pm 0.00$ |
| | | vanilla | $0.30 \pm 0.03$ | $\mathbf{8.50 \pm 0.06}$ | $\mathbf{11.50 \pm 0.00}$ | $\mathbf{10.92 \pm 0.02}$ | $0.98 \pm 0.00$ | $0.78 \pm 0.00$ | $0.93 \pm 0.03$ |
| | FREED | filtered | $0.47 \pm 0.20$ | $6.18 \pm 1.03$ | $9.00 \pm 0.36$ | $8.42 \pm 0.31$ | $0.98 \pm 0.00$ | $0.81 \pm 0.06$ | $\mathbf{0.99 \pm 0.00}$ |
| | | vanilla | $0.44 \pm 0.32$ | $5.71 \pm 0.92$ | $8.60 \pm 0.91$ | $8.00 \pm 0.69$ | $0.89 \pm 0.03$ | $0.55 \pm 0.09$ | $0.67 \pm 0.38$ |
| abl1 | FREED++ | filtered | $0.33 \pm 0.07$ | $\mathbf{11.16 \pm 0.09}$ | $\mathbf{13.13 \pm 0.37}$ | $12.33 \pm 0.19$ | $\mathbf{1.00 \pm 0.00}$ | $0.73 \pm 0.18$ | $0.99 \pm 0.00$ |
| | | vanilla | $0.34 \pm 0.02$ | $11.00 \pm 0.10$ | $12.86 \pm 0.05$ | $\mathbf{12.36 \pm 0.21}$ | $0.99 \pm 0.00$ | $0.73 \pm 0.08$ | $0.97 \pm 0.02$ |
| | FREED | filtered | $0.02 \pm 0.03$ | $2.92 \pm 4.50$ | $5.70 \pm 5.06$ | $5.12 \pm 4.45$ | $\mathbf{1.00 \pm 0.00}$ | $0.60 \pm 0.53$ | $\mathbf{1.00 \pm 0.00}$ |
| | | vanilla | $0.24 \pm 0.23$ | $2.17 \pm 1.68$ | $7.83 \pm 3.75$ | $6.36 \pm 5.28$ | $0.92 \pm 0.07$ | $\mathbf{0.83 \pm 0.16}$ | $0.89 \pm 0.14$ |
| usp7 | FREED++ | filtered | $0.64 \pm 0.07$ | $\mathbf{11.37 \pm 0.23}$ | $13.76 \pm 0.40$ | $13.27 \pm 0.23$ | $0.99 \pm 0.00$ | $0.92 \pm 0.03$ | $0.99 \pm 0.00$ |
| | | vanilla | $0.64 \pm 0.07$ | $9.66 \pm 3.28$ | $\mathbf{14.16 \pm 0.47}$ | $\mathbf{13.30 \pm 0.13}$ | $0.94 \pm 0.04$ | $0.91 \pm 0.04$ | $0.99 \pm 0.00$ |
| | FREED | filtered | $0.14 \pm 0.23$ | $3.22 \pm 5.14$ | $5.13 \pm 6.76$ | $4.05 \pm 6.57$ | $\mathbf{1.00 \pm 0.00}$ | $\mathbf{0.99 \pm 0.01}$ | $\mathbf{1.00 \pm 0.00}$ |
| | | vanilla | $0.10 \pm 0.07$ | $8.58 \pm 1.02$ | $11.36 \pm 0.98$ | $11.12 \pm 0.88$ | $0.99 \pm 0.00$ | $0.90 \pm 0.06$ | $0.93 \pm 0.06$ |

Table 11: Effect of fragment library filtration for FREED++ and FREED.

**Results**   The metrics are shown in table 11. Using a filtered fragments library results in similar docking score metrics during generation. However, filtering the fragments library increases the proportion of molecules that pass chemical filters. Since the filtration preprocessing step does not require additional computational resources and generally improves the generation quality, we conclude that it is helpful to include it in the generation pipeline.

The same effect can be seen for FREED: filtration procedure increases the percentage of valid molecules in terms of considered filters. As in previous experiments with FREED, it commonly tends to diverge. Because of original implementation issues analyzing filtration impact on DS metrics is difficult.

## H   Reward computation

Following FREED paper, we estimate binding affinity with OpenBabel and QuickVina2 tools.  First, we generate initial molecular conformation with OpenBabel gen3d operation, which consists of several steps: the generation of an initial approximation using a predefined set of rules and 3D templates, energy optimization with an empirical forcefield, the search for a conformation with low energy in rotor space with a genetic algorithm, and the energy optimization with conjugate gradient method. Further, the optimal pose and energy of the protein-ligand complex are searched with QuickVina2. Conceptually it uses a form of Memetic Algorithm, which interleaves the Monte Carlo algorithm with a restart for global search and BFGS method for local optimization. QuickVina2 provides a parameter called exhaustiveness, which defines the computational effort for docking simulation. Simulations with high exhaustiveness sample more candidates, which improves the performance of search, but requires more computational time. Following FREED we use

exhaustiveness=1 during training to save computational time. During evaluation (see sec. 5) we generate 3 different initial conformations with OpenBabel, dock them with QuickVina2 with exhaustiveness=8, and take the minimum over obtained estimates, to obtain a reliable estimate of binding affinity.

## I  USP7 structure

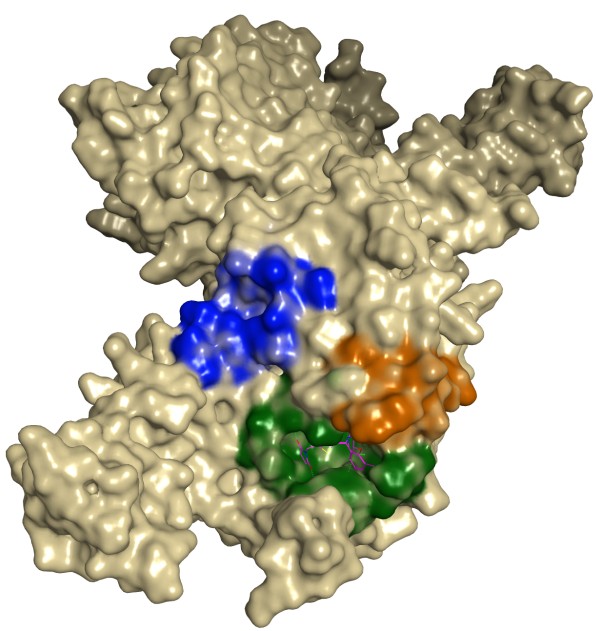

Figure 9: Structure of USP7 catalytic domain in complex with molecule 23 (PDB ID: 6VN3). The molecular surface was added. The orange surface depicts the catalytic center; the green surface depicts the cleft between the Thumb and Palm domains; the blue surface depicts the allosteric binding site. The molecule 23 is shown by stick model inside the cleft.

## J  Time Limits

Following FREED, we start generation with a benzene ring with three attachment points and restrict ourselves to 4 assembling steps. After the molecule is assembled, we perform a docking simulation and assign a reward for the episode. Hence we state generation as a time-limited task. Note that we define the state as a partially assembled molecule, described by extended molecular graphs (see sec. 3.2), ignoring any notion of time. This problem formulation results in 2 major drawbacks.

First, it needs to be clarified how to select a time limit properly. Authors of FREED motivate the choice of four assembling steps by the fact that active compounds from DUDE are most commonly fragmented into 4-6 blocks. However, the number of fragments in a molecule can vary significantly depending on the protein of interest (see figure 10).

Second, in the described setup, the agent faces a state aliasing problem (Whitehead & Ballard, 1991) that arises when an extended graph can be assembled from several sets of fragments of different sizes. As described in (Pardo et al., 2018), the time-unaware agent cannot accurately estimate the value function because an estimate for the same state will differ depending on whether the time limit is reached.

One possible solution is to include a notion of the remaining time as part of the agent's input (Pardo et al., 2018), which was utilized in MolDQN; however, it is still unclear how to select a time limit. A more promising solution would be the controllable generation termination when the termination signal is predicted with an auxiliary neural network based on the current state representation, similar to the one used in GCPN and Pocket2Mol. We leave this for future work.

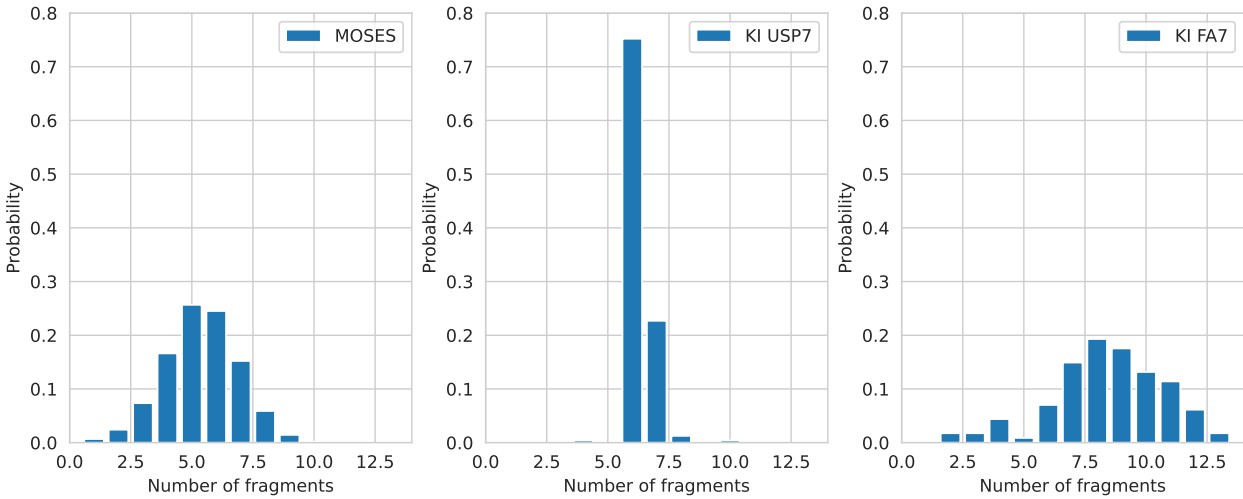

Figure 10: Distribution of a number of fragments in molecules according to BRICS fragmentation. KI stands for "Known Inhibitors"."

## K    Major implementation issues

In this section, we evaluate the significance of each major implementation issue from Section 4.1. First, we take the FFREED and add bugs one at a time (see Table 12). Then, we take the original FREED and remove bugs one at a time (see Table 13).

| protein | method | unique (↑) | Avg DS (↑) | Max DS (↑) | Top-5 DS (↑) |
|---|---|---|---|---|---|
| 5ht1b | FFREED* | $0.45 \pm 0.16$ | $\mathbf{11.97 \pm 0.20}$ | $\mathbf{14.85 \pm 0.21}$ | $\mathbf{13.99 \pm 0.45}$ |
| | FFREED(CRITIC_ARC) | $0.50 \pm 0.08$ | $10.25 \pm 2.14$ | $14.46 \pm 0.50$ | $13.44 \pm 0.44$ |
| | FFREED(CRITIC_UPD) | $0.62 \pm 0.20$ | $11.03 \pm 0.55$ | $14.33 \pm 0.45$ | $13.24 \pm 0.17$ |
| | FFREED(GUMBEL) | $1.00 \pm 0.00$ | $7.82 \pm 1.35$ | $12.70 \pm 0.26$ | $11.21 \pm 0.05$ |
| | FFREED(TARGET_NET)* | $0.43 \pm 0.13$ | $11.46 \pm 0.19$ | $14.70 \pm 0.42$ | $13.63 \pm 0.20$ |
| usp7 | FFREED | $0.66 \pm 0.01$ | $\mathbf{10.91 \pm 0.15}$ | $\mathbf{14.03 \pm 0.46}$ | $\mathbf{13.35 \pm 0.27}$ |
| | FFREED(CRITIC_ARC) | $0.66 \pm 0.06$ | $9.34 \pm 0.21$ | $13.50 \pm 0.62$ | $12.58 \pm 0.25$ |
| | FFREED(CRITIC_UPD) | $0.46 \pm 0.24$ | $8.91 \pm 0.07$ | $13.80 \pm 0.60$ | $12.84 \pm 0.11$ |
| | FFREED(GUMBEL) | $1.00 \pm 0.00$ | $7.58 \pm 1.36$ | $11.96 \pm 0.50$ | $10.64 \pm 0.19$ |
| | FFREED(TARGET_NET) | $0.52 \pm 0.06$ | $10.11 \pm 0.57$ | $14.03 \pm 0.45$ | $13.28 \pm 0.23$ |
| akt1 | FFREED | $0.44 \pm 0.05$ | $\mathbf{8.87 \pm 0.24}$ | $\mathbf{10.66 \pm 0.40}$ | $\mathbf{10.17 \pm 0.17}$ |
| | FFREED(CRITIC_ARC) | $0.60 \pm 0.13$ | $8.09 \pm 0.91$ | $10.20 \pm 0.34$ | $9.64 \pm 0.21$ |
| | FFREED(CRITIC_UPD) | $0.51 \pm 0.04$ | $8.58 \pm 0.16$ | $10.36 \pm 0.20$ | $9.78 \pm 0.23$ |
| | FFREED(GUMBEL) | $1.00 \pm 0.00$ | $4.96 \pm 2.62$ | $9.96 \pm 0.73$ | $8.60 \pm 0.26$ |
| | FFREED(TARGET_NET) | $0.55 \pm 0.19$ | $8.38 \pm 0.06$ | $10.03 \pm 0.25$ | $9.58 \pm 0.07$ |

Table 12: Adding bugs to FFREED. Asterix (*) denotes models with seeds with a positive Avg DS during training but negative Max DS on evaluation. We exclude such runs.

The main takeaway from Table 12 is that adding all major bugs leads to performance degradation. The Gumbel-Softmax issue is the most significant one. Next, we take the original FREED and remove bugs one at a time (see Table 13).

| protein | method | unique (↑) | Avg DS (↑) | Max DS (↑) | Top-5 DS (↑) |
|---------|--------|------------|------------|------------|--------------|
| 5ht1b | FREED | 0.21 ± 0.21 | 6.43 ± 5.57 | 8.83 ± 7.65 | 7.97 ± 6.90 |
|  | FREED(CRITIC_ARC)* | 0.00 ± 0.00 | 4.95 ± 7.00 | 4.95 ± 7.00 | 4.95 ± 7.00 |
|  | FREED(CRITIC_UPD) | 0.19 ± 0.33 | 4.66 ± 4.13 | 6.43 ± 6.08 | 5.78 ± 5.61 |
|  | FREED(GUMBEL) | 0.98 ± 0.02 | **7.41 ± 2.48** | **13.40 ± 0.55** | **12.16 ± 0.19** |
|  | FREED(TARGET_NET) | 0.67 ± 0.44 | 6.78 ± 1.11 | 12.56 ± 0.77 | 11.74 ± 0.38 |
| usp7 | FREED | 0.10 ± 0.07 | **8.58 ± 1.02** | 11.36 ± 0.98 | 11.12 ± 0.88 |
|  | FREED(CRITIC_ARC) | 0.06 ± 0.05 | 6.19 ± 1.84 | 10.76 ± 2.48 | 10.37 ± 2.14 |
|  | FREED(CRITIC_UPD) | 0.27 ± 0.46 | 6.84 ± 3.94 | 10.50 ± 1.17 | 7.71 ± 4.65 |
|  | FREED(GUMBEL) | 0.99 ± 0.00 | 7.71 ± 1.37 | 12.30 ± 0.39 | 11.33 ± 0.34 |
|  | FREED(TARGET_NET) | 0.82 ± 0.26 | 8.35 ± 0.22 | **12.86 ± 0.30** | **11.41 ± 0.56** |
| akt1 | FREED | 0.70 ± 0.45 | 3.98 ± 0.21 | 9.26 ± 0.30 | 8.47 ± 0.26 |
|  | FREED(CRITIC_ARC) | 0.30 ± 0.52 | 3.62 ± 2.72 | 6.33 ± 4.06 | 5.35 ± 3.92 |
|  | FREED(CRITIC_UPD) | 0.14 ± 0.20 | 1.30 ± 2.25 | 3.06 ± 5.31 | 2.89 ± 5.01 |
|  | FREED(GUMBEL) | 0.99 ± 0.00 | 5.29 ± 1.38 | **9.83 ± 0.41** | **8.77 ± 0.07** |
|  | FREED(TARGET_NET) | 1.00 ± 0.00 | **5.84 ± 0.97** | 9.73 ± 0.23 | 8.73 ± 0.13 |

Table 13: Removing bugs from FREED. Asterix (*) denotes models with seeds with a positive Avg DS during training but negative Max DS on evaluation. We exclude such runs.

This ablation shows that removing any particular bug does not lead to significant performance improvements. All the bugs need to be fixed simultaneously.

