# OpenReview forum: "FREED++: Improving RL Agents for Fragment-Based Molecule Generation by Thorough Reproduction"
_TMLR — Accepted by TMLR_

### Review · Reviewer_rFQy · 2023-10-12

**Summary Of Contributions:**

This paper studies FREED, which is a reinforcement learning-based molecular generation method. After introducing FREED, this paper points out bugs and issues of the original implementation of FREED and addresses these issues, resulting in the fixed version called FFREED. Moreover, this paper proposes to replace a fusing function for fragment embeddings with a simplified function and called the modified version FREED++. The performance of FREED, FFREED, and FREED++ is empirically examined on real-world molecular generation benchmark datasets.

**Audience:**

No

**Broader Impact Concerns:**

I do not have any concerns.

**Claims And Evidence:**

No

**Requested Changes:**

Please resolve all the above weaknesses.

**Strengths And Weaknesses:**

### Strengths

- The performance of the fixed FREED, FFREED, is superior to the original FREED, hence it can be practically valuable.
- The proposed FREED++ is more efficient than FREED while keeping the competitive (or superior) performance, hence it can be also valuable contribution for TMLR's audience.
### Weaknesses

1. The effectiveness of one of the main contributions of this paper, FFREED, is restricted to the original implementation of FREED, which significantly limits the audience  of this paper.
3. The procedure of message passing is discussed as one of major issues in Section 4.1, where the authors mention that graphs are treated as directed in the original implementation and it is not appropriate. However, if a graph is directed by definition in a dataset, considering the direction in message passing should be more appropriate than treating it as an undirected graph. So it is not correct to always treat graphs as undirected and this point should be more carefully discussed.
4. More detailed empirical evaluation would be desirable. In FFREED, there are five major modifications as stated in Section 4.1. Thus each of them should be empirically evaluated separately via ablation study.
5. In experiments, I recommend using the GuacaMol benchmark, which has been commonly used in molecular generation.
6. Presentation should be improved.
	1. Since the authors mention about the "original" implementation of FREED, I guess the authors discuss about https://github.com/AITRICS/FREED , which has been provided in the original FREED paper. However, it is better to be explicitly refer the link in the paper.
	2. The section title of Section 3 is "FREED++", but it seems that Section 3 just introduces the existing FREED. So the title should be modified as "FREED".
	3. Similarly, in the beginning of Section 3, there is a sentence "This section provides a detailed description of the proposed FREED++ method." However, this section is not about the proposal but the existing FREED method. So this should be also revised. If there is some delta that is not contained in the original FREED in this section, it should be clarified further.
	4. I do not fully understand the "Fragment selection" improvement in FREED++ in Section 4.3 because the process discussed here and Equation (4) do not have one-to-one correspondence. Please clarify this point.

---

> ### Author Response · Authors · 2023-10-27
> **Answer to reviewer rFQy (1/2)**
>
> ## Weaknesses
>
> **Weakness 1**
> > The effectiveness of one of the main contributions of this paper, FFREED, is restricted to the original implementation of FREED, which significantly limits the audience of this paper.
>
> We understand your concern regarding the lack of novelty and the limited audience of the paper. However, we have specifically chosen the TMLR journal as it encourages the authors to submit papers which contain “reproducibility studies of previously published results or claims.”
> The primary purpose of our work is to show that careful reproduction and examination of previously published research can significantly improve the quality of models. Our fixed model FREED++ achieves a new SOTA in the protein-conditioned generation task, which we consider a valuable contribution to the field and believe to be interesting to the TMLR audience.
>
> **Weakness 2**
> > The procedure of message passing is discussed as one of the major issues in Section 4.1, where the authors mention that graphs are treated as directed in the original implementation and it is not appropriate. However, if a graph is directed by definition in a dataset, considering the direction in message passing should be more appropriate than treating it as an undirected graph. So it is not correct to always treat graphs as undirected and this point should be more carefully discussed.
>
> We agree that some graphs are directed by design and that the directed message passing is appropriate for such data. However, most previous works on molecular property prediction [1] and drug discovery [2] consider molecular graphs undirected. For this reason, we initially considered the directed message passing to be a major implementation issue. However, after conducting an additional ablation experiment (the results are in the table below) with directed and undirected molecular graphs in FREED++, we moved the “message passing” paragraph to section 4.2.
>
> | protein | name | unique | Avg DS | Max DS | Top-5 DS |
> |---|---|---|---|---|---|
> | parp1 | FREED++ | $0.71 \pm 0.06$ | $12.98 \pm 0.15$ | $\mathbf{17.73 \pm 0.83}$ | $\mathbf{16.26 \pm 0.19}$ |
> | parp1 | FREED++(DIR) | $0.68 \pm 0.08$ | $\mathbf{13.59 \pm 0.06}$ | $17.13 \pm 0.86$ | $15.80 \pm 0.28$ |
> | 5ht1b | FREED++ | $0.45 \pm 0.07$ | $\mathbf{11.78 \pm 0.12}$ | $\mathbf{14.50 \pm 0.79}$ | $\mathbf{13.50 \pm 0.15}$ |
> | 5ht1b | FREED++(DIR) | $0.54 \pm 0.22$ | 10.92 ± 0.15 | $13.73 \pm 0.20$ | $12.81 \pm 0.09$ |
>
> As can be seen from the table, the average DS for the FREED++ with the directed graph is similar to the one we report in the manuscript.
>
> [1] Wieder, O., Kohlbacher, S., Kuenemann, M., Garon, A., Ducrot, P., Seidel, T., & Langer, T. (2020). A compact review of molecular property prediction with graph neural networks. Drug Discovery Today: Technologies, 37, 1-12.
>
> [2] Sun, M., Zhao, S., Gilvary, C., Elemento, O., Zhou, J., & Wang, F. (2020). Graph convolutional networks for computational drug development and discovery. Briefings in bioinformatics, 21(3), 919-935.
>
> **Weakness 3**
> > More detailed empirical evaluation would be desirable. In FFREED, there are five major modifications as stated in Section 4.1. Thus each of them should be empirically evaluated separately via ablation study.
>
> We would like to stress that the changes in section 4.1 are not modifications. They describe bugs found in the code and corresponding fixes. Thus, we do not see a reason to evaluate each modification separately.
> That being said, following your and the reviewer’s PMcY suggestion, we conducted an additional ablation experiment with directed and undirected graphs in FREED++ (refer to weakness 2).
>
> **Weakness 4**
> > In experiments, I recommend using the GuacaMol benchmark, which has been commonly used in molecular generation.
>
> Thank you for your suggestion. We agree that additional benchmarking is useful and will consider it in future work. However, to the best of our knowledge, this particular benchmark does not contain any ligand generation tasks.

---

> > ### Author Response · Authors · 2023-10-27
> > **Answer to reviewer rFQy (2/2)**
> >
> > **Weakness 5**
> > > Since the authors mention about the “original” implementation of FREED, I guess the authors discuss about https://github.com/AITRICS/FREED , which has been provided in the original FREED paper. However, it is better to be explicitly refer the link in the paper.
> >
> > We agree that it is necessary to reference the original implementation explicitly. We added the link in the revision.
> >
> > > The section title of Section 3 is “FREED++”, but it seems that Section 3 just introduces the existing FREED. So the title should be modified as “FREED”. Similarly, in the beginning of Section 3, there is a sentence “This section provides a detailed description of the proposed FREED++ method.” However, this section is not about the proposal but the existing FREED method. So this should be also revised. If there is some delta that is not contained in the original FREED in this section, it should be clarified further.
> >
> > Section 3 describes the “FREED++” framework. FREED++ and FREED share a lot in common. However, in the FREED paper, multiple essential framework details were not included. We believe that a thorough description of these subtleties is important for the understanding of FREED++. Therefore, we describe the full architecture in our manuscript. Following your suggestion, we added a sentence explaining the structure of the paper in the first paragraph of section 3.
> >
> > > I do not fully understand the “Fragment selection” improvement in FREED++ in Section 4.3 because the process discussed here and Equation (4) do not have one-to-one correspondence. Please clarify this point.
> >
> > Equation 4 describes the fragment selection process of FREED++, whereas section 4.3 describes the fragment selection process of the original FREED.

---

> > > ### Comment · Reviewer_rFQy · 2023-11-09
> > >
> > > Thank you for your reply and revising the paper.
> > >
> > > **Weakness 1**
> > >
> > > I am not saying about the novelty, and I understand the scope of TMLR. What I am worried about is that, since the bugs are specific issues of the original implementation, if someone publishes another implementation of FREED that resolves all the bugs at some point, the value of this paper could significantly decrease.
> > >
> > > **Weakness 2**
> > >
> > > The revised sentence in Section 4.2 is well written. Thank you for addressing this concern.
> > > I think "threating" of "... that threating molecules as undirected graphs ..." in the last sentence should be "treating".
> > >
> > > **Weakness 3**
> > >
> > > I apologize for my confusing wording. I understand it and "modifications" in my sentence actually means fixes of bugs.
> > > And I still think that it is important to empirically evaluate the effect of each bug fix via ablation studies to see which bug is crucial and which bug is minor.
> > >
> > > **Weakness 4**
> > >
> > > I thought that this paper considers not only ligand generation but the general molecular generation task.
> > > If this paper is specific about the ligand generation task, this should be clearly stated in the paper (maybe in title/abst/intro).
> > > Otherwise experiments on Guacamol could be valuable.
> > >
> > > **Weakness 5**
> > >
> > > > Section 3 describes the “FREED++” framework. FREED++ and FREED share a lot in common. However, in the FREED paper, multiple essential framework details were not included. We believe that a thorough description of these subtleties is important for the understanding of FREED++.
> > >
> > > I am still a bit confused. If the "multiple essential framework details" are already included in FREED but just not clearly written in the original paper, I still believe that the section title should be FREED.
> > > What an important issue is to clearly distinguish (1) components in FREED, (2) components in FREED but not clearly explained in the original paper, and (3) components not in FREED and newly added in FREED++. The section title should be FREED++ if (3) is written, but otherwise it should be just FREED.
> > >
> > > > Equation 4 describes the fragment selection process of FREED++, whereas section 4.3 describes the fragment selection process of the original FREED.
> > >
> > > I have already understood it. What I was saying is that, since the definition of $p_2$ in the original paper is not described in the paper (am I missing?), it is hard to see the difference between procedures of FREED and FREED++. I think it is helpful to write down both formulae in Sec 4.2 so that the reader can directly compare them.

---

> ### Author Response · Authors · 2023-11-10
> **Reply to the official comment by reviewer rFQy**
>
> Thank you for engaging in the discussion! It helps us improve the manuscript significantly.
>
> **Weakness 1**
>
> First, we would like to stress that our contribution is not limited to the fixed version of FREED (FFREED). We also propose simplifications that speed up the training by a factor of 3 and reduce the number of trainable parameters by a factor of 6 while maintaining similar performance (see Tables 9 and 10 in the manuscript). Besides that, we significantly extend the experimental evaluation of the model: we test FREED, FFREED, and FREED++ against several SOTA protein-conditioned generation approaches (see Section 5.1), study the effect of different fragmentation techniques and databases (see Section 5.2), and show that FREED++ is capable of generating ligands with high Docking Score in a real-world scenario (see Section 5.3).
>
> **Weakness 2**
>
> Thank you for pointing this out! We will fix this typo (along with other grammatical errors in this section) in the next revision.
>
> **Weakness 3**
>
> Thank you for the clarification of your idea. We agree that it may provide additional insights. Therefore, we propose the following two-sided ablation. First, we will select three protein targets: 1 out of 3 proteins studied in the original paper, 1 out of 10 proteins shown in Figure 7, and the USP7 protein. Then, we will add major bugs from section 4.1 to FFREED one at a time and compare the performance of the bugged models to the FFREED. Finally, we will take the original implementation of FREED and remove major bugs from section 4.1 one at a time to compare these models to the FREED. We hope this experiment will sufficiently show the effect of major implementation issues on the performance. Unfortunately, we cannot ablate minor implementation issues due to limited computational resources. What are your thoughts on this experiment? Are there any specific modifications you would like us to ablate?
>
> **Weakness 4**
>
>  We would like to note that, in theory, the FREED model can solve the general molecular generation task if provided with the proper reward function. However, we only experimented with the target protein generation task, as it is the main focus of the original FREED paper. We will add a clarifying sentence to the intro in the next revision.
>
> **Weakness 5**
>
> We agree that the current layout of the paper can be confusing. However, we believe that including another section that describes the original model and its implementation will substantially increase the length of the manuscript while bringing even more confusion. As a workaround, we propose the following, we will highlight all the places in section 3 that are different from the original paper and provide additional descriptions of the dissimilarities. That way, we will be able to constrain the size of the manuscript and provide readers with a clear explanation of the modifications.

---

> > ### Comment · Reviewer_rFQy · 2023-11-12
> >
> > Thank you for your quick reply.
> > The proposed revision will address my concerns, and I think the paper will be valuable in the community.

---

> > > ### Author Response · Authors · 2023-12-01
> > > **Reply to the  official comment by reviewer rFQy**
> > >
> > > We have performed the described experiment. We dub the bugs described in Section 4.1 (in the order of appearance) as “CRITIC_ARC,” “CRITIC_UPD,” “GUMBEL,” and “TARGET_NET.” First, we report the results of the experiment in which we add major bugs from section 4.1 to FFREED one at a time:
> > >
> > > | protein | method | unique ($\uparrow$) | Avg DS ($\uparrow$) | Max DS ($\uparrow$) | Top-5 DS ($\uparrow$) |
> > > | --- | --- | --- | --- | --- | --- |
> > > |  | FFREED* | 0.45 ± 0.16 | **11.97 ± 0.20** | **14.85 ± 0.21** | **13.99 ± 0.45** |
> > > |  | FFREED(CRITIC_ARC) | 0.50 ± 0.08 | 10.25 ± 2.14 | 14.46 ± 0.50 | 13.44 ± 0.44 |
> > > | 5ht1b | FFREED(CRITIC_UPD) | 0.62 ± 0.20 | 11.03 ± 0.55 | 14.33 ± 0.45 | 13.24 ± 0.17 |
> > > |  | FFREED(GUMBEL) | 1.00 ± 0.00 | 7.82 ± 1.35 | 12.70 ± 0.26 | 11.21 ± 0.05 |
> > > |  | FFREED(TARGET_NET)* | 0.43 ± 0.13 | 11.46 ± 0.19 | 14.70 ± 0.42 | 13.63 ± 0.20 |
> > > |  |  |  |  |  |  |
> > > |  | FFREED | 0.66 ± 0.01 | **10.91 ± 0.15** | **14.03 ± 0.46** | **13.35 ± 0.27** |
> > > |  | FFREED(CRITIC_ARC) | 0.66 ± 0.06 | 9.34 ± 0.21 | 13.50 ± 0.62 | 12.58 ± 0.25 |
> > > | usp7 | FFREED(CRITIC_UPD) | 0.46 ± 0.24 | 8.91 ± 0.07 | 13.80 ± 0.60 | 12.84 ± 0.11 |
> > > |  | FFREED(GUMBEL) | 1.00 ± 0.00 | 7.58 ± 1.36 | 11.96 ± 0.50 | 10.64 ± 0.19 |
> > > |  | FFREED(TARGET_NET) | 0.52 ± 0.06 | 10.11 ± 0.57 | 14.03 ± 0.45 | 13.28 ± 0.23 |
> > > |  |  |  |  |  |  |
> > > |  | FFREED | 0.44 ± 0.05 | **8.87 ± 0.24** | **10.66 ± 0.40** | **10.17 ± 0.17** |
> > > |  | FFREED(CRITIC_ARC) | 0.60 ± 0.13 | 8.09 ± 0.91 | 10.20 ± 0.34 | 9.64 ± 0.21 |
> > > | akt1 | FFREED(CRITIC_UPD) | 0.51 ± 0.04 | 8.58 ± 0.16 | 10.36 ± 0.20 | 9.78 ± 0.23 |
> > > |  | FFREED(GUMBEL) | 1.00 ± 0.00 | 4.96 ± 2.62 | 9.96 ± 0.73 | 8.60 ± 0.26 |
> > > |  | FFREED(TARGET_NET) | 0.55 ± 0.19 | 8.38 ± 0.06 | 10.03 ± 0.25 | 9.58 ± 0.07 |
> > >
> > > The table’s main takeaway is that adding any major bug leads to performance degradation. The Gumbel-Softmax issue is the most significant one. Next, we take the original FREED and remove bugs one at a time.
> > >
> > > | protein | method | unique ($\uparrow$) | Avg DS ($\uparrow$) | Max DS ($\uparrow$) | Top-5 DS ($\uparrow$) |
> > > | :--- | :--- | :---: | :---: | :---: | :---: |
> > > |  | FREED | 0.21 ± 0.21 | 6.43 ± 5.57 | 8.83 ± 7.65 | 7.97 ± 6.90 |
> > > |  | FREED(CRITIC_ARC)* | 0.00 ± 0.00 | 4.95 ± 7.00 | 4.95 ± 7.00 | 4.95 ± 7.00 |
> > > | 5ht1b | FREED(CRITIC_UPD) | 0.19 ± 0.33 | 4.66 ± 4.13 | 6.43 ± 6.08 | 5.78 ± 5.61 |
> > > |  | FREED(GUMBEL) | 0.98 ± 0.02 | **7.41 ± 2.48** | **13.40 ± 0.55** | **12.16 ± 0.19** |
> > > |  | FREED(TARGET_NET) | 0.67 ± 0.44 | 6.78 ± 1.11 | 12.56 ± 0.77 | 11.74 ± 0.38 |
> > > |  |  |  |  |  |  |
> > > |  | FREED | 0.10 ± 0.07 | **8.58 ± 1.02** | 11.36 ± 0.98 | 11.12 ± 0.88 |
> > > |  | FREED(CRITIC_ARC) | 0.06 ± 0.05 | 6.19 ± 1.84 | 10.76 ± 2.48 | 10.37 ± 2.14 |
> > > | usp7 | FREED(CRITIC_UPD) | 0.27 ± 0.46 | 6.84 ± 3.94 | 10.50 ± 1.17 | 7.71 ± 4.65 |
> > > |  | FREED(GUMBEL) | 0.99 ± 0.00 | 7.71 ± 1.37 | 12.30 ± 0.39 | 11.33 ± 0.34 |
> > > |  | FREED(TARGET_NET) | 0.82 ± 0.26 | 8.35 ± 0.22 | **12.86 ± 0.30** | **11.41 ± 0.56** |
> > > |  |  |  |  |  |  |
> > > |  | FREED | 0.70 ± 0.45 | 3.98 ± 0.21 | 9.26 ± 0.30 | 8.47 ± 0.26 |
> > > |  | FREED(CRITIC_ARC) | 0.30 ± 0.52 | 3.62 ± 2.72 | 6.33 ± 4.06 | 5.35 ± 3.92 |
> > > | akt1 | FREED(CRITIC_UPD) | 0.14 ± 0.20 | 1.30 ± 2.25 | 3.06 ± 5.31 | 2.89 ± 5.01 |
> > > |  | FREED(GUMBEL) | 0.99 ± 0.00 | 5.29 ± 1.38 | **9.83 ± 0.41** | **8.77 ± 0.07** |
> > > |  | FREED(TARGET_NET) | 1.00 ± 0.00 | **5.84 ± 0.97** | 9.73 ± 0.23 | 8.73 ± 0.13 |
> > >
> > > This ablation shows that removing any particular bug does not lead to significant performance improvements. All the bugs need to be fixed simultaneously.

---

### Review · Reviewer_PMcY · 2023-10-13

**Summary Of Contributions:**

The authors benchmark the recently published FREED algorithm, providing additional experimental details and improvements. The authors then compare their improved version, FFREED and FREED++, to additional benchmark algorithms and tasks.

**Audience:**

Yes

**Claims And Evidence:**

Yes

**Requested Changes:**

“Despite being cheaper and faster than the traditional wet-lab approach, VS has another major drawback: it is limited to the databases of already known drug candidates and is not suitable for de novo drug design.” Why is this true? Why cannot novel molecular candidates be used for CADD?

“It uses a modification of the DQN algorithm…” - Please don’t use acronym

“Research that reproduces and improves previous work” - I don’t think this section is necessary. I don’t think you need to justify review and critique of previously published work, especially if you are making improvements therein.

Section header: “3.1 MDP” - This could have a much better title. What do you mean by MDP? It isn’t defined anywhere else in the work.

In Figure 4, what is the null distribution of Tanimoto similarity? What would the similarity be at random, when using that same fragment library and assembly rules? I.e. What is the null distribution?

Terminology:

Section 3.1:
\overline{DS(s_{t+1})} = -DS(s_{t+1})  : Why not just keep it negative? This makes it look like the average, which is confusing.

What are the units for the Docking Score?

Section 3.3 : What is MI? I think this is good to define, especially because you refer to it later in section 4.3.

Section 3.4 : “The first scenario involves applying the critic to the actions saved in the replay buffer.” Can you better describe the replay buffer? Why is it called that?

Section 4.1 : Can the “target network” be defined before when discussing the FREED algorithm? It is unclear what the motivation of such a network is. If it wasn’t used in the original algorithm, did it have a large effect when the right hyperparameter was chosen?

Fusing function : What are z and x? Have these variable been defined anywhere yet, and how do they refer to either the action or the state? Second, isn’t the concatenation layer, and the associated matrix Q, equivalent to simply removing the W learnable tensor and using just U and V?

**Strengths And Weaknesses:**

Strengths

Generally, I think it is good to publish when another algorithm is lacking, and explain why it does or doesn’t work. I think the reimplementation is convincing and helpful to show their contribution.

Interesting to break down the two different kind of RL-based algorithms: sequential vs pretrained generative.

I think the “Current state of protein-conditioned molecule generation” is thorough and up-to-date.

Writing out the steps in the action is very helpful.

The description of the different fragment library collections is helpful.

Weaknesses

The message passage section in 4.1 is confusing. “The direction of the edge is determined by the inner representation of the graph in RDKit” – what does this mean? Moreover, in practice, does updating that part of the network improve performance? If so, where do you show this?

For the USP7 molecule generation task, how many different ligands are available, and what is the diversity of molecule poses? It’d be great to see how well this framework generalizes across a wide variety of ligands and binding pocket poses. Are docking scores validated by in vitro binding?

---

> ### Author Response · Authors · 2023-10-27
> **Answer to reviewer PMcY (1/3)**
>
> ## Weaknesses
>
> **Weakness 1**
> > The message passage section in 4.1 is confusing. “The direction of the edge is determined by the inner representation of the graph in RDKit” – what does this mean? Moreover, in practice, does updating that part of the network improve performance? If so, where do you show this?
>
> In the original implementation, the adjacency matrix of the molecular graph is built from the corresponding adjacency list. Adjacency lists computed with “GetBonds” method only include each pair of vertices once, so the resulting graph appears to be directed. In theory, such a procedure can lead to a graph with isolated vertices (no edges pointing). The embeddings of such vertices do not get updated in the message passing phase and thus cannot aggregate information about other vertices in the graph. These isolated vertices are why most implementations of GNNs for molecules [1, 2] use the undirected representation of molecular graphs.
>
> Despite that, and because of concerns expressed by you and reviewer rFQy, we conducted an additional experiment on two protein targets and compared FREED++ with directed and undirected graphs. The results are in the table below.
>
> | protein | name | unique | Avg DS | Max DS | Top-5 DS |
> |---|---|---|---|---|---|
> | parp1 | FREED++ | $0.71 \pm 0.06$ | $12.98 \pm 0.15$ | $\mathbf{17.73 \pm 0.83}$ | $\mathbf{16.26 \pm 0.19}$ |
> | parp1 | FREED++(DIR) | $0.68 \pm 0.08$ | $\mathbf{13.59 \pm 0.06}$ | $17.13 \pm 0.86$ | $15.80 \pm 0.28$ |
> | 5ht1b | FREED++ | $0.45 \pm 0.07$ | $\mathbf{11.78 \pm 0.12}$ | $\mathbf{14.50 \pm 0.79}$ | $\mathbf{13.50 \pm 0.15}$ |
> | 5ht1b | FREED++(DIR) | $0.54 \pm 0.22$ | 10.92 ± 0.15 | $13.73 \pm 0.20$ | $12.81 \pm 0.09$ |
>
> As can be seen from the table, the average DS for the FREED++ with the directed graph is similar to the one we report in the manuscript. Considering this result, we moved the message passing paragraph from section 4.1 (major implementation issues) to section 4.2 (minor implementation issues) as we still consider it an issue due to the problem described earlier.
>
> [1] Wieder, O., Kohlbacher, S., Kuenemann, M., Garon, A., Ducrot, P., Seidel, T., & Langer, T. (2020). A compact review of molecular property prediction with graph neural networks. Drug Discovery Today: Technologies, 37, 1-12.
>
> [2] Sun, M., Zhao, S., Gilvary, C., Elemento, O., Zhou, J., & Wang, F. (2020). Graph convolutional networks for computational drug development and discovery. Briefings in bioinformatics, 21(3), 919-935.
>
> **Weakness 2**
> > For the USP7 molecule generation task, how many different ligands are available, and what is the diversity of molecule poses? It’d be great to see how well this framework generalizes across a wide variety of ligands and binding pocket poses. Are docking scores validated by in vitro binding?
>
> The paper on USP7 [1] describes 40 compounds and studies their ability to bind to the protein. Figures 2 and 3 in [1] show two significantly different binding poses of different compounds. The IC_50 metric is reported for all the compounds (see Tables 1-6 in [1]). We do not validate the docking scores for the molecules generated by FREED++. As our work is meant to showcase the general potential of the fragment-based drug design approach, the complete development of the inhibitor for a specific protein, which includes the in vitro validation, is out of the scope of our paper.
> Experimental results in Table 6 and Figure 7 in our work illustrate that our model generalizes well to different protein targets. We did not experiment with different binding pockets of the same protein but expect the same level of performance. We do not think that generalization to a “variety of ligands” is a well-defined question, as our model is trained to generate new ligands.

---

> > ### Author Response · Authors · 2023-10-27
> > **Answer to reviewer PMcY (2/3)**
> >
> > ## Requested changes
> >
> > **Change 1**
> > > “Despite being cheaper and faster than the traditional wet-lab approach, VS has another major drawback: it is limited to the databases of already known drug candidates and is not suitable for de novo drug design.” Why is this true? Why cannot novel molecular candidates be used for CADD?
> >
> > The virtual screening approach uses a discriminative model to determine the qualities of already known substances. It does not provide any means of generating new substances. That being said, it can be paired with a generative model (i.e., combinatorial generator) to determine the qualities of new substances. In this case, however, its performance will be entirely determined by the generator.
> >
> > **Change 2**
> > > “It uses a modification of the DQN algorithm…” - Please don’t use acronym
> >
> >  DQN is an RL model introduced in [1]. It stands for “Deep Q-Network”. We replaced the acronym with the full name in the revision of the manuscript.
> >
> > [1] Mnih, V., Kavukcuoglu, K., Silver, D., Rusu, A. A., Veness, J., Bellemare, M. G., ... & Hassabis, D. (2015). Human-level control through deep reinforcement learning. nature, 518(7540), 529-533.
> >
> > **Change 3**
> > > “Research that reproduces and improves previous work” - I don’t think this section is necessary. I don’t think you need to justify review and critique of previously published work, especially if you are making improvements therein.
> >
> > We agree that the section “Research that reproduces and improves previous work” is unnecessary and remove it from the manuscript.
> >
> > **Change 4**
> > > Section header: “3.1 MDP” - This could have a much better title. What do you mean by MDP? It isn’t defined anywhere else in the work.
> >
> > MDP is a commonly used in RL acronym. It stands for the “Markov Decision ‘Process” and defines the environment the RL agent interacts with. Refer to [1] for further details.
> >
> > [1] Sutton, R. S., & Barto, A. G. (2018). Reinforcement learning: An introduction. MIT press.
> >
> > **Change 5**
> > > In Figure 4, what is the null distribution of Tanimoto similarity? What would the similarity be at random, when using that same fragment library and assembly rules? I.e. What is the null distribution?
> >
> > To answer this question, we collected 1000 molecules with a uniform policy using different fragment libraries and measured the maximal Tanimoto similarity to the set of known inhibitors for each molecule. In the table below, we report the mean maximal Tanimoto similarity over the set of generated molecules and its standard deviation.
> >
> > | Fragment library | Mean-max Tanimoto similarity |
> > |---|---|
> > | BRICS-MOSES | $0.70 \pm 0.07$ |
> > | BRICS-USP7 | $0.52 \pm 0.11$ |
> > | CREM-ZINC | $0.42 \pm 0.07$ |
> >
> > Surprisingly, the most similar to known inhibitor molecules are obtained when using the “BRICS-MOSES” fragment library.
> >
> > ### Terminology
> > **Change 6**
> > > Section 3.1: \overline{DS(s_{t+1})} = -DS(s_{t+1}) : Why not just keep it negative? This makes it look like the average, which is confusing.
> >
> > We agree that the usage of \overline{DS} may be confusing and remove it from the manuscript. We thank you for the valuable suggestion and redefine the DS in the manuscript as the negative binding energy estimated by the docking software. We added an additional explanatory sentence in section 3.1.
> >
> > **Change 7**
> > > What are the units for the Docking Score?
> > The Docking Score is commonly measured in KCal/Mol. We also report DS values in KCal/Mol. We added this information in the revision.
> >
> > **Change 8**
> > > Section 3.3 : What is MI? I think this is good to define, especially because you refer to it later in section 4.3.
> >
> > MI stands for Multiplicative Interactions. We agree that the usage of the acronym is misleading and fix it in the revision.
> >
> > **Change 9**
> > > Section 3.4 : “The first scenario involves applying the critic to the actions saved in the replay buffer.” Can you better describe the replay buffer? Why is it called that?
> >
> > The usage of the replay buffer is a common practice in off-policy RL [1, 2, 3]. Experience obtained by the actor while interacting with the environment is stored in the replay buffer and used in the training stage.
> >
> > [1] Mnih, V., Kavukcuoglu, K., Silver, D., Rusu, A. A., Veness, J., Bellemare, M. G., ... & Hassabis, D. (2015). Human-level control through deep reinforcement learning. nature, 518(7540), 529-533.
> >
> > [2] Fujimoto, S., Hoof, H., & Meger, D. (2018, July). Addressing function approximation error in actor-critic methods. In International conference on machine learning (pp. 1587-1596). PMLR.
> >
> > [3] Haarnoja, T., Zhou, A., Abbeel, P., & Levine, S. (2018, July). Soft actor-critic: Off-policy maximum entropy deep reinforcement learning with a stochastic actor. In International conference on machine learning (pp. 1861-1870). PMLR.

---

> > > ### Author Response · Authors · 2023-10-27
> > > **Answer to reviewer PMcY (3/3)**
> > >
> > > **Change 10**
> > > > Section 4.1 : Can the “target network” be defined before when discussing the FREED algorithm? It is unclear what the motivation of such a network is. If it wasn’t used in the original algorithm, did it have a large effect when the right hyperparameter was chosen?
> > >
> > > Target networks are widely used in Reinforcement Learning [1, 2, 3] to stabilize the training of the Q-function. Target networks aim to solve the moving target problem that arises in temporal-difference learning due to bootstrapping.
> > >
> > > [1] Mnih, V., Kavukcuoglu, K., Silver, D., Rusu, A. A., Veness, J., Bellemare, M. G., ... & Hassabis, D. (2015). Human-level control through deep reinforcement learning. nature, 518(7540), 529-533.
> > >
> > > [2] Fujimoto, S., Hoof, H., & Meger, D. (2018, July). Addressing function approximation error in actor-critic methods. In International conference on machine learning (pp. 1587-1596). PMLR.
> > >
> > > [3] Haarnoja, T., Zhou, A., Abbeel, P., & Levine, S. (2018, July). Soft actor-critic: Off-policy maximum entropy deep reinforcement learning with a stochastic actor. In International conference on machine learning (pp. 1861-1870). PMLR.
> > >
> > > **Change 11**
> > > > Fusing function : What are z and x? Have these variable been defined anywhere yet, and how do they refer to either the action or the state? Second, isn’t the concatenation layer, and the associated matrix Q, equivalent to simply removing the W learnable tensor and using just U and V?
> > >
> > > We call a “Fusing function" a function aggregating two different sources of information (tensors) into a single tensor. In equation 10, we define the “Multiplicative Interaction” fusing function introduced in [1]. This function was used in the original FREED implementation. In equation 11, we define the concatenation function, which is a go-to fusing function in DL. Variables $z, x$ denote input embedding tensors to be aggregated by the fusing function. Depending on the action step, they can denote the aggregated graph embedding of the molecule and the embedding tensor of attachment points (equation 2); the embedding of the selected attachment points and the embedding of the selected fragment (equation 6); the fused embedding of the selected attachment point and the selected fragment and the embedding tensor of the attachment points on the selected fragment (equation 7).
> > > Yes, a multiplicative interaction without the $\mathbf{W}$ learnable tensor is equivalent to concatenation fusing.
> > >
> > > [1] Jayakumar, S. M., Czarnecki, W. M., Menick, J., Schwarz, J., Rae, J., Osindero, S., ... & Pascanu, R. (2020). Multiplicative interactions and where to find them.

---

### Review · Reviewer_4UNS · 2023-10-17

**Summary Of Contributions:**

The paper introduces FREED++, which is a re-implementation of the FREED algorithm for fragment based molecular generation. The authors claim that the original implementation had various bug and issues and suggest additional improvements that are captured in FREED++, including:
* Updates and critic architecture and training
* Using undirected graphs for message

The authors run various experiments on FREED++, as well as the original baselines in the FREED paper, and generally find that their of FFREED and FREED++ outperform the original FREED. The papers also shows additional results on the effects of fragment creation through CReM and BRICS and a case study for the design of USP7 inhibitors.

**Audience:**

Yes

**Broader Impact Concerns:**

No major concerns

**Claims And Evidence:**

Yes

**Requested Changes:**

Please address the following to increase chances for acceptance:
* Clarify if the experimental results (other than FFREED and FREED++) are run based on your implementation or taken from other papers.
* Provide greater clarity on your contributions and why they would be interesting to TMLR readers.
   * I think the contributions could be further strengthened by running additional experiments that change network architectures (e.g. different GNNs) or training mechanisms (e.g. different RL algorithms).
* Place the reward function definitions for each experiment in the main paper.
* Add additional related work:
   * RL for molecular design using text-based methods [1] [2]
   * Other methods for molecular design in graph-based settings, such GFlowNets [3] [4]

[1] Raj Ghugare and Santiago Miret and Adriana Hugessen and Mariano Phielipp and Glen Berseth. "Searching for High-Value Molecules Using Reinforcement Learning and Transformers." arXiv preprint arXiv:2310.02902 (2023).
[2] Mazuz, E., Shtar, G., Shapira, B., & Rokach, L. (2023). Molecule generation using transformers and policy gradient reinforcement learning. Scientific Reports, 13(1), 8799.
[3] Bengio, E., Jain, M., Korablyov, M., Precup, D., & Bengio, Y. (2021). Flow network based generative models for non-iterative diverse candidate generation. Advances in Neural Information Processing Systems, 34, 27381-27394.
[4] Jain, M., Raparthy, S. C., Hernández-Garcı́a, A., Rector-Brooks, J., Bengio, Y., Miret, S., & Bengio, E. (2023, July). Multi-objective gflownets. In International Conference on Machine Learning (pp. 14631-14653). PMLR.

**Strengths And Weaknesses:**

Strengths:
* The paper provides significant detail on describing various parts of the problem setting, for both the original FREED implementation and the improvements made in FREED++.
* The paper provides additional experiments in creating fragment and USP7 inhibitors.

Weaknesses:
* It is unclear of the the original FREED is an implementation of the authors. It would be good to clarify since the authors claim that the original implementation contains bugs.
* The papers spend a lot real estate explaining the prior method (FREED) and less on the new contributions. I think the paper would be stronger by focusing more of the new contributions and why they would be of interest to TMLR readers.
* Some important experimental details, such as the reward functions for all the experiments, are only briefly described in the appendix. It would be good to have them defined in the main part of the paper.

---

> ### Author Response · Authors · 2023-10-27
> **Answer to reviewer 4UNS**
>
> ## Weaknesses
>
> **Weakness 1**
>  > It is unclear of the the original FREED is an implementation of the authors. It would be good to clarify since the authors claim that the original implementation contains bugs.
>
> We agree that the absence of a reference to the original implementation is misleading. We use the implementation listed in the original FREED paper: https://github.com/AITRICS/FREED. We added the reference in the revision.
>
> **Weakness 2**
> > The papers spend a lot real estate explaining the prior method (FREED) and less on the new contributions. I think the paper would be stronger by focusing more of the new contributions and why they would be of interest to TMLR readers.
>
> We understand the concern regarding the lack of novelty in the paper. However, we have specifically chosen the TMLR journal as it encourages the authors to submit papers that contain “reproducibility studies of previously published results or claims.”
> The primary purpose of our work is to show that careful reproduction and examination of previously published research can significantly improve the quality of models. Our fixed model FREED++ achieves a new SOTA in the protein-conditioned generation task, which we consider a valuable contribution to the field and believe to be interesting to the TMLR audience.
> The paper spends so much time on FREED as it is only possible to explain the implementation issues and modifications with an in-depth description of the original work.
>
> **Weakness 3**
> > Some important experimental details, such as the reward functions for all the experiments, are only briefly described in the appendix. It would be good to have them defined in the main part of the paper.
>
> Our work mainly focuses on the Docking Score (DS) optimization. The optimized reward is described in section 3.1. This reward function is used for FREED and FREED++ in all the experiments except for the “Development of USP7 inhibitors” (see section 5.3). The reward functions for REINVENT and MolDQN are described in section 5.1.
>
> ## Requested changes
>
> **Change 1**
> > Clarify if the experimental results (other than FFREED and FREED++) are run based on your implementation or taken from other papers.
>
> We train the REINVENT model from scratch using the official implementation (https://github.com/MarcusOlivecrona/REINVENT) with the Docking score as the scoring function. Moreover, we perform a grid search for the $\sigma$ hyperparameter. For MolDQN, we also use the official implementation (https://github.com/aksub99/MolDQN-pytorch) and train all the models from scratch using the Docking score as the optimization objective. Furthermore, for MolDQN, we test two setups: the first is identical to the original implementation with dense rewards and high-frequency updates; the second uses sparse rewards and low-frequency updates, as described in section 5.1. For Pocket2Mol, we used a pretrained model provided by the authors (https://github.com/pengxingang/Pocket2Mol). We used the implementation from MOSES (https://github.com/molecularsets/moses) for the combinatorial generator (CombGen). Finally, we trained all baseline FREED models for all proteins using the official implementation (https://github.com/AITRICS/FREED).
> Following your request, we added links to all implementations to the revision of the paper.
>
> **Change 2.1**
> > Provide greater clarity on your contributions and why they would be interesting to TMLR readers
>
> Please see the response to weakness 2.
>
> **Change 2.2**
> > I think the contributions could be further strengthened by running additional experiments that change network architectures (e.g. different GNNs) or training mechanisms (e.g. different RL algorithms).
>
> Thank you for pointing out an exciting direction for the future work. However, in this work, we focus on careful reproduction and simplification of the original FREED and feel that the additional experiments with different GNN architectures and RL algorithms are out of the scope of the paper.
>
> **Change 3**
> > Place the reward function definitions for each experiment in the main paper.
>
> Please see the response to weakness 3.
>
> **Change 4**
> > Add additional related work.
>
> Thank you for the suggestion. We added all the mentioned papers in the revision of the manuscript.

---

> > ### Comment · Reviewer_4UNS · 2023-11-18
> > **Thank you for the updates**
> >
> > Thank you for the updates to the draft and addressing my comments.

---

### Author Response · Authors · 2023-12-01
**General Comment**

We would like to thank all the reviewers for engaging in the discussion! It helped us significantly improve the paper. We have uploaded a new revision of the manuscript. Changes made since the last revision are highlighted in blue. The main changes are listed below:
1. We provided a link to the source code.
1. We cited the GuacaMol [1] benchmark in the introduction section.
2. The presentation in Section 3 has been improved. We highlighted the differences between FREED and FREED++ with blue boxes and fixed the inconsistency between Figure 3b and Section 3.4.
3. We performed an additional ablation experiment to evaluate the significance of each particular major implementation issue. The results are in the Appendix.
4. Fixed typos and grammatical errors.

[1] Brown, N., Fiscato, M., Segler, M. H., & Vaucher, A. C. (2019). GuacaMol: benchmarking models for de novo molecular design. Journal of chemical information and modeling, 59(3), 1096-1108.

---

### Comment · Action_Editors · 2023-12-29
**Camera ready version request**

Dear Authors,

The camera-ready version states "Code and Reproducibility
We will happily provide the code and the scripts that reproduce all the experiments from this manuscript
in the camera-ready version.". Could you please replace this with the link to the scripts?

Thank you,
AE

---

> ### Author Response · Authors · 2023-12-29
>
> Dear Action Editor,
>
> Thank you for pointing out the inconsistency in the "Code and Reproducibility" section! Along with the corrected revision, we provide source code and configs for all methods as supplementary material.

---

### Decision · Action_Editor_bXPP · 2023-12-03

**Recommendation:** Accept as is

**Comment:**

The paper is a thorough study of reproducing and finally improving FREED, a fragment-based method for molecule generation. Reviewers have questioned the novelty of the work and raised the concern that the potential audience might be limited to those interested specifically in the algorithm. However, the Authors show through a thorough experimental evaluation a strong empirical performance, which suggests that the results are likely to be relevant as a reference point for other molecule generation methods. In the end, all reviewers were supportive of accepting the paper. The paper also serves as an important reminder that implementation details are extremely important to get right. Thank you for submitting your work to TMLR. All in all, it is my pleasure to recommend accepting the paper as is.

**Audience:**

Reviewers have questioned the novelty of the work and raised the concern that the potential audience might be limited to those interested specifically in the algorithm. However, the Authors show through a thorough experimental evaluation a strong empirical performance, which suggests that the results are likely to be relevant as a reference point for other molecule generation methods.

**Claims And Evidence:**

The paper presents a very thorough experimental study of FREED, a fragment-based method for molecule generation. The results are supported by clear evidence.